# Reduction of Probabilistic Chemical Reaction Networks

**Mauricio Montes** [* 1 2]   **Gregoire Sergeant-Perthuis** [* 2]

## Abstract

Programming adaptive behaviors at the cellular level is a long-standing goal that raises the question of how probabilistic computation can be implemented in biochemical systems. Chemical reaction networks (CRNs) provide such a substrate and have been shown to realize probabilistic models, including hidden Markov models and factor graphs, with dynamics reproducing Bayesian inference and belief propagation. However, encoding these algorithms typically requires prohibitively large reaction networks, and classical CRN reduction techniques do not directly apply. By recovering the factor graph structure encoded in Napp–Adams-compiled CRNs, we transport recent factor-graph reduction results to their chemical implementations, obtaining significantly smaller CRNs while preserving the belief-propagation fixed points on surviving variables.

## 1. Introduction

Models of agency in biological organisms, such as the Bayesian brain hypothesis (Knill & Pouget, 2004) and active inference (Friston, 2010; Parr et al., 2022), posit that decision making relies on maintaining and updating internal beliefs about the environment. Underlying these frameworks is the assumption that biological agents operate under incomplete information, which they acquire only indirectly through noisy and partial observations. As a consequence, adaptive behavior requires continuous inference about latent environmental states.

**Modeling agency *in silico*.** Incomplete information is also central to optimal control and, more broadly, to reinforcement learning, where belief updates are formalized through probabilistic inference algorithms such as filtering and belief propagation. These same inference mechanisms also

play a central role in active inference (Pezzulo et al., 2024). Translating these algorithms into biological substrates therefore appears as a necessary step toward cellular decision making design. A natural choice for achieving this goal is using chemical reaction networks (CRNs), denoted $\Gamma$, which are a list of reactions whose mass-action kinetics define an ordinary differential equation (ODE) for concentrations of its species, providing a continuous-time substrate for computation. Recent work has shown that CRNs can implement a range of probabilistic inference tasks: Bayesian posterior computation in algal phototaxis (Colliaux et al., 2017), filtering (Nakamura & Kobayashi, 2022; 2021), belief propagation (Napp & Adams, 2013), inference (Poole et al., 2017), learning hidden Markov models (Wiuf et al., 2023), and classification (Moghimianavval et al., 2024; van der Linden et al., 2022). Information processing via chemical regulation is imperative to the creation of composable biochemical systems that can generalize to different tasks (Cardelli et al., 2018; Brijder, 2018; Tschantz et al., 2019; Vasic et al., 2021; Viswa Virinchi et al., 2018). However, implementing these tasks with CRNs scales poorly as the number of distinct chemical species (and associated reactions/ODEs) increase, that is, as decision-making problems become more complex (Napp & Adams, 2013; Hjelmfelt et al., 1991; Viswa Virinchi et al., 2018; Wiuf et al., 2023; Poole et al., 2017).

**Inference with CRNs.** In this article, we focus on inference in factor graphs, denoted by $\mathcal{F}$, which subsume main ingredients of the methods and models discussed above (Anderson et al., 2021; Napp & Adams, 2013; Anderson et al., 2021; Vasic et al., 2021; Lakin, 2023). Engineering biological systems to scale up to large sizes is an essential task at the intersection of these fields. Just as users do not need detailed knowledge of the intermediate steps of the internal calculations being done by a computer, being able to simplify and interpret designed biochemical systems to the inputs and outputs of interest is necessary to reduce the burden of control in the design of programmable cells (Tschantz et al., 2019). Limitations to these approaches arise because such implementations typically rely on the control of the stoichiometry of a synthetic CRN. To that end, we leverage recent factor graph reduction techniques based on deformation retractions of topological spaces associated to factor graphs (Sergeant-Perthuis & Boitel, 2025) and

---

[1]COSAM, Auburn University, Auburn, AL, USA [2]CQSB, Sorbonne Universite, Paris, France. Correspondence to: Mauricio Montes <mauricio.montes@auburn.edu>.

*Proceedings of the 43rd International Conference on Machine Learning*, Seoul, South Korea. PMLR 306, 2026. Copyright 2026 by the author(s).

transport these reductions to their chemical implementation.

**Contributions** We develop a theory for recognizing and reducing CRNs that implement belief propagation via the Napp–Adams construction (Napp & Adams, 2013). The key insight is that these CRNs contain catalytic dependencies that are not visible from stoichiometry alone: messages are represented by species bundles, factor tables appear as rate parameters, and BP fixed points are encoded by bundle-normalized steady states. We make four contributions: (1) structural conditions (W1–W6) under which a mass-action CRN encodes a factor graph, together with a constructive reconstruction procedure (Theorem 4.7); (2) a proof that positive steady states correspond bijectively to BP fixed points under an additional recycling condition R1 (Theorem 5.4); (3) a functorial transport of SP–B factor-graph retractions to CRN reductions that delete entire message bundles while preserving BP fixed points on surviving variables (Proposition 6.2, Lemma 6.3) — to our knowledge the first semantics-preserving reduction for BP-compiled CRNs; and (4) empirical validation across several graph families (Section 7). The scope of (1–3) is intentional: W1–W6 characterize the Napp–Adams compiled subclass, not generic biochemical networks, and the pipeline is closed under reduction by Proposition 6.2. Table 2 (Appendix A) summarizes the correspondence between graphical and chemical concepts that (W1–W6) formalize. We recommend consulting it while reading Sections 4–6.

The invariant preserved here differs from classical CRN reductions. Stoichiometric and Laplacian-based reductions preserve structural or steady-state properties visible from reaction stoichiometry. In Napp–Adams compiled CRNs, however, the BP message passing algorithm is carried by catalytic dependencies and bundle-normalized concentration ratios. These dependencies are not captured by stoichiometry alone, which is why we reduce at the factor-graph level and then transport the reduction to the CRN.

## 2. Main Results

We study a class of probabilistic CRNs: mass-action networks whose positive steady states encode belief propagation (BP) fixed points, which give approximate marginals for an associated factor graph, reconstructed from the CRN's message-bundle structure. Our results formalize a pipeline that (1) compiles a factor graph into a probabilistic CRN, (2) recognizes and reconstructs the underlying factor graph from a given CRN, and (3) transports factor-graph reductions to the chemical setting in a way that preserves BP solutions.

**Compilation (Factor graphs → CRNs).** Napp and Adams (Napp & Adams, 2013) show how to implement

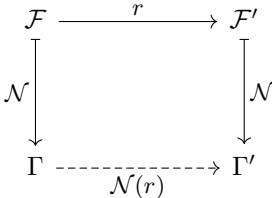

*Figure 1.* An SP–B retraction $r$ at the factor-graph level induces a CRN reduction $\mathcal{N}(r)$ that preserves BP fixed points on surviving variables (Proposition 6.2, Lemma 6.3).

message-passing inference in a CRN by introducing, for each message vector, a bundle of species whose relative concentrations represent that vector. Reactions are chosen so that the resulting mass-action ODE performs a continuous-time analogue of damped asynchronous message updates, converging to the same fixed points. In our paper we treat this construction as a compilation map from a factor graph $\mathcal{F}$ to a CRN $\Gamma$ (formal definitions appear in Section 6 and Appendix D).

**Recognition and reconstruction (CRNs → Factor graphs).** We give conditions (W1)–(W6) under which a CRN encodes a factor graph together with its factor tables, and we provide an explicit reconstruction procedure $\mathrm{FG}(\Gamma)$ (Theorem 4.7).

**Steady states and BP fixed points.** With one additional condition (R1) ensuring the correct normalization/recycling behavior, we prove that positive steady states of $\Gamma$ correspond to BP fixed points on $\mathrm{FG}(\Gamma)$ (Theorem 5.4).

**Reduction commutes with compilation.** Sergeant–Perthuis and Boitel (SP–B) (Sergeant-Perthuis & Boitel, 2025) introduce a notion of *retraction* for factor graphs, obtained by deleting certain variable/factor nodes and updating the neighboring factor tables in a way that preserves the set of BP fixed points on the remaining variables. We package these SP–B retractions as morphisms in our factor-graph category (Section 6). We then show that the Napp–Adams compilation is functorial: an SP–B retraction $\mathcal{F} \to \mathcal{F}'$ induces a corresponding CRN reduction $\Gamma \to \Gamma'$ that removes entire *message bundles* and updates only the rate parameters in the affected local subnetworks (Proposition 6.2). Consequently, compiling after reducing yields the same reduced CRN (up to relabeling) as reducing after compiling, and BP fixed points are preserved under this transport (Lemma 6.3).

**Empirical gains.** Across our benchmark suite, SP–B retractions translate into large *chemical* savings after Napp–Adams compilation, and these savings show up directly as faster ODE integration. Figure 2 illustrates the typi-

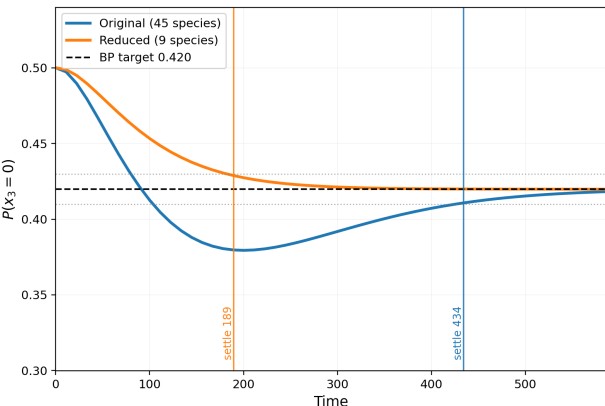

*Figure 2.* **Simulation Improvements** SP–B reductions substantially shrink compiled CRNs and yield large compilation and simulation-time speedups on our benchmark suite while preserving inferred marginals on surviving variables. Here we have an $80\%$ reduction the size of the CRN (3-chain) and a $56\%$ speed up in convergence to the correct marginal (Section 7).

cal behavior: retracting reducible substructure substantially shrinks the compiled CRN and accelerates convergence of the concentration-derived marginals. Table 1 summarizes results by graph family. Tree- and chain-structured instances exhibit near-complete collapse to a small core (about $95\%$ fewer variables and $97\%$ fewer chemical species at the median), yielding dramatic simulation speedups (median $270\times$ on chains and $886\times$ on trees). Loopy-core instances with reducible tendrils see partial but still substantial compression (median $79\%$ variable and $73\%$ species reduction) and corresponding speedups (median $22\times$), reflecting that the irreducible loopy core must remain. By contrast, grid graphs are essentially irreducible under these moves and show no change in size or runtime, serving as a control. Finally, random planted-core graphs admit only modest reduction (median $20\%$ variables and $10\%$ species) and therefore modest speedups (median $1.5\times$), consistent with having fewer retractable tendrils. In all cases, we additionally monitor marginal agreement on the surviving variables (Section 7); the observed runtime gains coincide with preserved inferred marginals rather than a change in the underlying beliefs. Section 7 also reports Erdős–Rényi stress tests over sparse, near-threshold, and dense regimes, showing that reduction tracks the amount of graph lying outside the irreducible loopy core rather than depending on a planted construction.

## 3. Background

**Factor graphs.** We consider discrete probabilistic models whose joint distribution is a product of local functions. A *factor graph* makes this factorization explicit and provides a convenient interface for message-passing inference.

**Definition 3.1** (Factor graph (Yedidia et al., 2000))**.** A factor graph $\mathcal{F} = (\mathcal{H}, E, f)$ consists of a hypergraph $\mathcal{H} = (I, J)$

*Table 1.* Median compression and simulation speedup of a structured benchmark suite.

| FAMILY | $n$ | VAR↓ (%) | SPECIES↓ (%) | SIM × |
|---|---|---|---|---|
| • CHAIN | 5 | 95.0 | 97.0 | 270.3 |
| • TREE | 4 | 95.1 | 97.5 | 885.5 |
| • LOOPY | 12 | 79.2 | 73.2 | 21.6 |
| • GRID | 4 | 0.0 | 0.0 | 1.0 |
| • RANDOM | 3 | 20.0 | 10.3 | 1.5 |

with variable indices $I$ and factor indices $J$, a collection of finite state spaces $(E_i)_{i \in I}$, and nonnegative factors $\psi_j : \prod_{i \in j} E_i \to \mathbb{R}_{\geq 0}$ for each $j \in J$.

The factors define a global probability distribution by

$$p(\mathbf{x}) = \frac{1}{Z} \prod_{j \in J} \psi_j(x_j), \qquad x_j := (x_i)_{i \in j},$$

where $Z$ is the normalizing constant. We depict $\mathcal{F}$ as a bipartite graph with variable nodes $i \in I$ and factor nodes $j \in J$, with an edge between $j$ and $i$ whenever $i \in j$.

**Belief propagation.** Belief propagation (BP) is a message-passing dynamic program that computes exact marginals on tree-structured factor graphs and is widely used as an approximation on graphs with cycles (loopy BP) (Yedidia et al., 2000). BP maintains messages along edges: variable-to-factor messages $P^{(i \to j)} \in \mathbb{R}_{\geq 0}^{E_i}$ and factor-to-variable messages $S^{(j \to i)} \in \mathbb{R}_{\geq 0}^{E_i}$. For $k \in E_i$,

$$S_k^{(j \to i)} = \sum_{x_{j \setminus i}} f_j(x_j) \prod_{i' \in j \setminus i} P_{x_{i'}}^{(i' \to j)},$$

$$P_k^{(i \to j)} = \prod_{j' \in \mathrm{ne}(i) \setminus j} S_k^{(j' \to i)}.$$

An (unnormalized) belief at variable $i$ is
$\tilde{p}(x_i = k) = \prod_{j \in \mathrm{ne}(i)} S_k^{(j \to i)}$.

**Chemical reaction networks.** A CRN is a finite set of species and reactions equipped with rate constants.

**Definition 3.2** (CRN)**.** A CRN is a triple $\Gamma = (X, R, \kappa)$ where $X$ is a finite set of species, $R$ is a finite set of reactions, and $\kappa : R \to \mathbb{R}_{>0}$ assigns a positive rate to each reaction.

Under mass-action kinetics, concentrations $x(t) \in \mathbb{R}_{\geq 0}^X$ evolve according to an ODE determined by $(X, R, \kappa)$. In this work, CRNs serve as a dynamical substrate for implementing and reducing probabilistic inference.

## 4. Recognition Conditions and Reconstruction

We introduce six local conditions on the CRN, (W1)–(W6), that can be checked directly in terms of stoichiometry and

rate polynomials. Together they guarantee that a factor-graph structure exists and can be reconstructed.

Before stating the recognition conditions, we briefly indicate their role. They abstract the bundle-and-rate pattern of the Napp–Adams compilation: messages become species bundles, factor tables become catalyzed production rates, and recycling supplies the normalization needed to recover BP fixed points at steady state. A fully worked three-variable chain example appears in Appendix E.1.

### 4.1. Stoichiometric conditions (W1–W3)

Let $\Gamma = (X, R, \kappa)$ be a CRN. A reaction $R_k \in R$ is expressed as $R_k : \sum_{i=1}^{N} r_{ik} X_i \longrightarrow \sum_{i=1}^{N} p_{ik} X_i$ with coefficients $r_{ik}, p_{ik} \in \mathbb{N}$. The *stoichiometric matrix* of $\Gamma$ is the $N \times K$ matrix $S = (x_{ik})$ with entries $x_{ik} = p_{ik} - r_{ik}$, so that column $k$ encodes the net change of species in reaction $R_k$.

**(W1) Stoichiometric block decomposition.** There exists a finite index set $\mathcal{B}$ and a partition of the set of species into disjoint pieces $S_B$

$$X = \bigsqcup_{B \in \mathcal{B}} S_B$$

such that, for each reaction $R_k$ with net stoichiometric column $(x_{ik})_{i=1}^{N}$, the support

$$\{\, i : x_{ik} \neq 0 \,\}$$

is contained in a single block $S_B$. In other words, every reaction has nonzero net stoichiometry in at most one block.

**(W2) Distinguished zero species in each block.** For each block $S_B$ in the partition of (W1) there exists a distinguished species $X_{B,0} \in S_B$, called the *zero species* of $S_B$. We require that $S_B \setminus \{X_{B,0}\}$ be nonempty and call its elements the *coordinate species* of $S_B$.

**(W3) Star–shaped internal stoichiometry.** For each block $S_B$ and each reaction $R_k$ whose net stoichiometry is supported in $S_B$, the restriction of the column $(x_{ik})$ to $S_B$ is either

$$-X_{B,0} + X_{B,k} \quad \text{or} \quad X_{B,0} - X_{B,k}.$$

Under (W1)–(W3), the species of $\Gamma$ split into disjoint blocks $S_B$, each with a unique zero species and several coordinate species, and all reaction columns are intra-block and of zero↔coordinate form.

*Remark* 4.1. If (W1)–(W3) hold, we refer to each block $S_B$ as a *bundle* and write

$$B = \{X_{B,0}\} \sqcup \{X_{B,k}\}_{k \in E_B},$$

where $E_B$ is the index set for the coordinate species of $B$. Condition (W3) then says that, within a bundle $B$, every reaction column $k$ has nonzero entries only in $B$ and its restriction to $B$ is either

$$-X_{B,0} + X_{B,k} \quad \text{or} \quad X_{B,0} - X_{B,k},$$

for a unique coordinate $X_{B,k} \in B \setminus \{X_{B,0}\}$.

**Definition 4.2** (Catalytic neighbors and assignments). Let $\Gamma = (X, R, \kappa)$ be a CRN with stoichiometric matrix $S$ satisfying (W1-W3).

1. For a bundle $B$, the *catalytic neighbor set* $\mathrm{Cat}(B)$ is the set of bundles $Q \neq B$ such that there exists a reaction $r \in R$ whose net stoichiometry is supported in $B$ and in which species of $Q$ appear with zero net stoichiometry.

2. For each $B$, write $E_Q$ for the coordinate index set of a bundle $Q \in \mathrm{Cat}(B)$, and define the *assignment set*

$$\mathcal{A}_B := \prod_{Q \in \mathrm{Cat}(B)} E_Q.$$

An element $a \in \mathcal{A}_B$ is a choice $a(Q) \in E_Q$ of one coordinate for each catalytic neighbor $Q \in \mathrm{Cat}(B)$.

### 4.2. Rate Conditions (W4–W6)

Let $B$ be a bundle with zero species $X_{B,0}$ and coordinate species $\{X_{B,k}\}_{k \in E_B}$, and let $\mathrm{Cat}(B)$ and $\mathcal{A}_B$ be as in Definition 4.2. For each $k \in E_B$, let $\frac{d}{dt}[X_{B,k}]^+$ denote the total *positive* contribution to $\frac{d}{dt}[X_{B,k}]$ coming from reactions whose net stoichiometry restricted to $B$ is $X_{B,0} \to X_{B,k}$.

**(W4) Separability of production rates.** We say that $\Gamma$ satisfies (W4) if for every bundle $B$ and every $k \in E_B$, the function $\frac{d}{dt}[X_{B,k}]^+$ is a polynomial in the concentrations of the catalytic neighbor coordinates, with *at most one* coordinate species from each catalytic neighbor bundle appearing in any monomial. That is, there exist coefficients $C_B(k; a) \geq 0$ indexed by $k \in E_B$ and assignments $a \in \mathcal{A}_B$ such that

$$\frac{d}{dt}[X_{B,k}]^+ = [X_{B,0}] \sum_{a \in \mathcal{A}_B} C_B(k; a) \prod_{Q \in \mathrm{Cat}(B)} [X_{Q,a(Q)}].$$

**Definition 4.3** (Sum-like vs product-like production). Assume (W1)–(W4) and let $C_B(k; a)$ be the assignment table of bundle $B$.

1. We say $B$ is *weakly product-like* if for every $k \in E_B$ there exists at most one assignment $a^{(k)} \in \mathcal{A}_B$ such that $C_B(k; a^{(k)}) > 0$, and $C_B(k; a) = 0$ for all $a \neq a^{(k)}$.

2. We say $B$ is *sum-like* if there exists some $k \in E_B$ and two distinct assignments $a \neq a'$ in $\mathcal{A}_B$ such that $C_B(k; a) > 0$ and $C_B(k; a') > 0$.

**Definition 4.4** (Assignment table). Suppose $\Gamma$ is a CRN satisfying (W1)–(W4). For each bundle $B$ and $k \in E_B$, write the positive production part $\frac{d}{dt}[X_{B,k}]^+$ as a polynomial in the catalytic coordinates $\{[X_{Q,i}]\}_{Q \in \text{Cat}(B), i \in E_Q}$. For each assignment $a \in \mathcal{A}_B$ let $C_B(k; a)$ be the coefficient of the monomial $[X_{B,0}] \prod_{Q \in \text{Cat}(B)} [X_{Q,a(Q)}]$.

**(W5) Coherent production for product-like bundles** Let $P$ be a bundle with $\text{Cat}(P) \neq \varnothing$ and assignment table $C_P(k; a)$ from (W4). We say that $P$ is *product-like* if the following conditions hold:

1. **(Unique supporting assignment)** For each $k \in E_P$, there exists **exactly one** assignment $a^{(k)} \in \mathcal{A}_P$ such that $C_P(k; a^{(k)}) > 0$, and $C_P(k; a) = 0$ for all $a \neq a^{(k)}$.

2. **(Diagonal structure via bijections)** For each catalytic neighbor $Q \in \text{Cat}(P)$, there exists a bijection $\theta_{P,Q} : E_P \to E_Q$ such that for all $k \in E_P$:
$$a^{(k)}(Q) = \theta_{P,Q}(k).$$

**(W6) Factor structure for sum-like bundles** Assume there exists a CRN satisfying (W1)–(W5). Condition (W6) requires that the sum-like bundles of $\Gamma$ can be partitioned into *factor classes*, each equipped with auxiliary data satisfying the following properties.

There exists a partition of the set of sum-like bundles into disjoint classes $\{J_1, J_2, \ldots\}$. For each factor class $J$, the **factor-class data** consists of:

- a finite set $V_J$ of product-like bundles (the *incident product bundles*),

- for each $Q \in V_J$, a bijection $\iota_{J \to Q} : E_Q \xrightarrow{\cong} E_{B_{J \to Q}}$ (the *state-space identification*),

- a strictly positive function $\psi_J : \prod_{Q \in V_J} E_Q \to (0, \infty)$ (the *factor table*),

- positive scalars $\{\kappa_{J \to Q} > 0\}_{Q \in V_J}$,

such that the following conditions hold:

**(W6.1) (Missing-one-neighbor pattern and targets)**

We have $|J| = |V_J|$, and for each $Q \in V_J$ there exists a unique bundle $B_{J \to Q} \in J$ with
$$\text{Cat}(B_{J \to Q}) = V_J \setminus \{Q\}.$$

This establishes a bijection between $J$ and $V_J$. We define the *target* of $B_{J \to Q}$ to be
$$\text{Tgt}(B_{J \to Q}) := Q,$$
and refer to $Q$ as the *target bundle* of $B_{J \to Q}$.

**(W6.2) (Target-compatible state spaces)**

For each $Q \in V_J$, the bijection $\iota_{J \to Q} : E_Q \to E_{B_{J \to Q}}$ witnesses that $|E_Q| = |E_{B_{J \to Q}}|$. Note that different $Q, Q' \in V_J$ may have $|E_Q| \neq |E_{Q'}|$, corresponding to variables of different cardinalities.

**(W6.3) (Clamped common table)**

For each $Q \in V_J$, each $k \in E_Q$, and each assignment $a \in \prod_{Q' \in V_J \setminus \{Q\}} E_{Q'}$, the assignment table of $B = B_{J \to Q}$ satisfies
$$C_B\big(\iota_{J \to Q}(k); a\big) =$$
$$\kappa_{J \to Q} \cdot \psi_J\big(x_Q = k, \ x_{Q'} = a(Q') \text{ for } Q' \neq Q\big).$$

Here $k \in E_Q$ is a state of the target product bundle, $\iota_{J \to Q}(k) \in E_B$ is the corresponding state in the sum bundle, and $a$ assigns states to the remaining product bundles in $V_J \setminus \{Q\}$.

**Definition 4.5** (Sum bundles and product bundles). Assume $\Gamma$ satisfies (W1)–(W6).

1. A *sum bundle* is any sum-like bundle, i.e., a bundle $B_{J \to Q}$ appearing in some factor class $(J, V_J)$.

2. A *product bundle* is any bundle $P$ that lies in $V_J$ for some factor class $(J, V_J)$—equivalently, $P$ is a catalytic neighbor of some sum bundle.

*Remark* 4.6 (Bundle extraction from W1–W3). Under (W1)–(W3) the species set $X$ decomposes into disjoint bundles $S_B$ with distinguished zero and coordinate species, and every internal reaction has the form $X_{B,0} \to X_{B,k}$ plus catalytic species from other bundles. In particular, the bundle structure and the classification into sum vs. product bundles can be recovered purely from stoichiometry and conserved quantities.

**Theorem 4.7** (CRN $\to$ Factor Graph Reconstruction). *Let $\Gamma = (X, R, \kappa)$ be a mass-action CRN satisfying* (W1)–(W6). *Then one can reconstruct a factor graph* $\text{FG}(\Gamma) = (V, E, H)$ *as follows:*

- *The factor nodes are the factor classes $J$ from* (W6).

- *The variable nodes are the variable classes $[P]$ under $\sim_v$ (Definition 5.1).*

- *A variable class $[P]$ is incident to a factor class $J$ if and only if $[P] \cap V_J \neq \varnothing$.*

- *The state space of variable class $[P]$ is $E_{[P]} := E_P$ for any representative $P \in [P]$.*

- *The factor table $\psi_J : \prod_{Q \in V_J} E_Q \to (0, \infty)$ from (W6) serves as the potential for factor node $J$.*

*Moreover, the incidence relation is well-defined: distinct elements of $V_J$ belong to distinct variable classes, establishing a bijection between $V_J$ and the variable classes incident to $J$. Each factor table $\psi_J$ is determined by $\Gamma$ up to multiplication by a positive constant.*

## 5. Reconstruction of the Belief Propagation Algorithm

### 5.1. Equivalence relations for factors and variables

**Definition 5.1** (Variable equivalence on product bundles)**.** Assume (W1)–(W6). Let $\mathcal{P}$ be the set of product bundles and $\mathcal{S}$ be the set of sum bundles.

Define a binary relation $\sim$ on $\mathcal{P}$ by: for $P, P' \in \mathcal{P}$, $P \sim P'$ if and only if:

$$\exists B \in \mathcal{S} \text{ such that: } \Big[ \text{Tgt}(B) = P \text{ and } B \in \text{Cat}(P') \Big]$$
$$\text{or } \Big[ \text{Tgt}(B) = P' \text{ and } B \in \text{Cat}(P) \Big].$$

Let $\sim_v$ be the equivalence relation on $\mathcal{P}$ generated by $\sim$ (the reflexive, symmetric and transitive closure).

The equivalence classes of $\mathcal{P}$ under $\sim_v$ are called *variable classes*.

**Lemma 5.2.** *The relation $\sim_v$ of Definition 5.1 is an equivalence relation on $\mathcal{P}$.*

**Lemma 5.3.** *If $P \sim_v P'$, then $|E_P| = |E_{P'}|$.*

Conditions (W1)–(W6) characterize when a CRN encodes a factor graph and its factor tables. To realize the Belief Propagation dynamics on the reconstructed factor graph, we impose one additional condition on the CRN:

**(R1) Bundlewise recycling and uniform product production.** For each bundle $B$ with zero species $X_{B,0}$ and coordinates $\{X_{B,k}\}_{k \in E_B}$ there exists a constant $\rho_B > 0$ such that, for every $k \in E_B$, the CRN contains an internal (non-catalyzed) recycling reaction

$$X_{B,k} \xrightarrow{\rho_B} X_{B,0},$$

and these are the only internal reactions whose net stoichiometry is $X_{B,k} \to X_{B,0}$.

Moreover, for each product bundle $P$, there exists a constant $\kappa_P > 0$ such that for every $k \in E_P$,

$$C_P(k; a^{(k)}) = \kappa_P,$$

where $a^{(k)}$ is the unique assignment supporting production of $P_k$ given by (W5). Equivalently, every producing reaction with internal net stoichiometry $P_0 \to P_k$ has rate constant $\kappa_P$, independent of $k \in E_P$.

**Theorem 5.4** (BP Implementation)**.** *Let $\Gamma = (X, R, \kappa)$ be a mass-action CRN satisfying (W1)–(W6) and (R1), and let $\text{FG}(\Gamma)$ be the factor graph with factor tables $\{\psi_J\}$ reconstructed by Theorem 4.7. Then the positive steady states of $\Gamma$ correspond bijectively to BP fixed points on $\text{FG}(\Gamma)$.*

*More precisely, at any positive steady state. For each factor class $J$ and target $Q \in V_J$, write $A_Q = \prod_{Q' \in V_J \setminus \{Q\}} E_{Q'}$ for the set of state assignments to the remaining variables, and write $\psi_J(k, a) := \psi_J\big(x_Q = k, x_{Q'} = a(Q') \text{ for all } Q' \in V_J \setminus \{Q\}\big)$ for the corresponding factor-table entry.*

1. *For each sum bundle $B = B_{J \to Q}$ associated to factor class $J$ and target product bundle $Q \in V_J$, the steady-state concentrations satisfy*

$$\frac{\rho_B}{[B_0]}[B_k] = \kappa_{J \to Q} \sum_{a \in A_Q} \psi_J(k, a) \prod_{Q' \in V_J \setminus \{Q\}} [Q'_{a(Q')}] \tag{1}$$

*for all $k \in E_Q$.*

2. *For each product bundle $P$, the steady-state concentrations satisfy*

$$\frac{\rho_P}{\kappa_P[P_0]}[P_k] = \prod_{B \in Cat(P)} [B_k] \tag{2}$$

*for all $k \in E_P$.*

3. *These equations are precisely the sum-product BP fixed-point equations on $\text{FG}(\Gamma)$, with the factors $\rho_B/[B_0]$ and $\rho_P/(\kappa_P[P_0])$ serving as the per-message normalization constants.*

## 6. Categories and the Compilation Functor

### 6.1. Source Category **FGraphRet**

Objects are pairs $(A, H)$, where $A$ is the poset associated to a factor graph and $H$ is the family of factor Hamiltonians. Morphisms are finite composites of admissible linear and colinear retractions in the sense of Sergeant–Perthuis–Boitel: these are order-preserving deformation retractions of $A$ which act on $H$ by:

- Deleting a variable $i$ and updating its incident factor by marginalizing out the corresponding coordinate in $H$ (linear point);

- Deleting a factor $j$ and incorporating its potential into neighboring factors (colinear point).

For the explicit formulas on $H$, see Propositions 5 and 6 of (Sergeant-Perthuis & Boitel, 2025). The explicit factor table updates (i.e. CRN coefficients) are provided in the Appendix D.1

## 6.2. Target Category CRN

Objects of **CRN** are mass-action reaction networks $\Gamma = (X, R, \kappa)$ satisfying the recognition conditions (W1)–(W6), where $X$ is the finite species set (partitioned into bundles), $R$ is the finite set of reactions, and $\kappa \in (0, \infty)^R$ is the vector of rate constants.

Morphisms in **CRN** are finite composites of elementary CRN reductions. An *elementary reduction* is a triple

$$(\varphi_0, \varphi_1, \psi) : (X, R, \kappa) \longrightarrow (X', R', \kappa')$$

with:

- $\varphi_0 : X \to X'$ obtained by deleting a union of whole bundles, so that $X' = X \setminus \bigcup_{B \in \mathcal{B}_{\mathrm{del}}} S_B$;

- $\varphi_1 : R \to R'$ the induced map on reactions, removing exactly those reactions that involve deleted species;

- $\psi : (0, \infty)^R \to (0, \infty)^{R'}$ a map on rate vectors, producing $\kappa' = \psi(\kappa)$ for the reduced network $(X', R')$.

A general morphism is a finite composite of such reductions, with composition defined componentwise:

$$(\varphi_0', \varphi_1', \psi') \circ (\varphi_0, \varphi_1, \psi) := (\varphi_0' \circ \varphi_0, \varphi_1' \circ \varphi_1, \psi' \circ \psi),$$

and identity morphisms given by $(\mathrm{id}_X, \mathrm{id}_R, \mathrm{id})$.

## 6.3. The Compilation Functor $\mathcal{N}$

We now package the Napp compilation into a functor

$$\mathcal{N} : \mathbf{FGraphRet} \longrightarrow \mathbf{CRN}.$$

**Action on objects.** Let $(A, H)$ be a factor graph in the sense of Sergeant–Perthuis–Boitel, with Hamiltonians $H_c$ and factor tables $\psi_c \propto e^{-H_c}$. The compiled CRN is

$$\mathcal{N}(A, H) = (X, R, \kappa),$$

constructed exactly as in Definition D.1.

**Action on morphisms.** Let

$$r : (A, H) \longrightarrow (A', H')$$

be a morphism in **FGraphRet**, i.e., a finite composite of linear and colinear retractions in the sense of Sergeant–Perthuis–Boitel. Write $\psi_c \propto e^{-H_c}$ and $\psi_c' \propto e^{-H_c'}$ for the initial and updated factor tables.

On CRNs, we define

$$\mathcal{N}(r) = (\varphi_0, \varphi_1, \psi) : \mathcal{N}(A, H) \to \mathcal{N}(A', H'),$$

as follows:

- $\varphi_0 : X \to X'$ deletes exactly the bundles corresponding to edges and variables removed by $r$ (linear or colinear point), and is the identity on all remaining species. At the level of bundles, $\varphi_0$ is just restriction of the Napp compilation to the surviving part of the factor graph.

- $\varphi_1 : R \to R'$ deletes all reactions whose reactants or products contain a deleted bundle; it is the identity on all other reactions (recycling, product, marginal, and surviving sum reactions).

- $\psi : \kappa \mapsto \kappa'$ acts only on the rate constants of surviving sum-bundle reactions. For each surviving factor node $j$ and edge $j \to n$, we keep the same scalar $c_{j \to n} > 0$ and set

$$\lambda_{j \to n}'(\mathbf{k}^j) = c_{j \to n} \cdot \psi_j'(\mathbf{k}^j),$$

where $\psi_j'$ is the updated factor from the SP–B retraction. Recycling and product rates are left unchanged:

$$\kappa_r' = \kappa_r, \quad \kappa_{\mathrm{prod}}' = \kappa_{\mathrm{prod}}.$$

By construction, $(\varphi_0, \varphi_1, \psi)$ is a morphism in **CRN**: it removes entire bundles and their reactions, and only rescales rate parameters on the surviving sum bundles, leaving the long-time behavior of the surviving species compatible with the factor-table update.

**Lemma 6.1** (Recycling structure of $\mathcal{N}(A, H)$). *For any factor graph $(A, H)$, the compiled CRN $\mathcal{N}(\mathcal{A}, H) = (X, R, \kappa)$ satisfies* (R1):

- *each bundle $B$ (sum, product, or marginal) contains a zero species $B_0$ and coordinate species $\{B_k\}_{k \in E_B}$;*

- *there is a single constant $\rho_B > 0$ such that, for every $k \in E_B$, the only internal reactions with net flow $B_k \to B_0$ are the recycling reactions*

$$B_k \xrightarrow{\rho_B} B_0,$$

*which involve no catalysts from other bundles.*

*In particular, the image of $\mathcal{N}$ lies in the full subcategory of CRNs satisfying* (W1)–(W6) *and* (R1).

*Proof of Lemma 6.1.* By construction of $\mathcal{N}(A, H)$, each bundle $B$ is equipped with recycling reactions

$$B_k \xrightarrow{\kappa_r} B_0 \quad \text{or} \quad B_k \xrightarrow{\kappa_{\mathrm{prod}}} B_0,$$

depending on whether $B$ is a message, belief, or marginal bundle, and no other reactions in $R$ have net stoichiometry supported in $\{B_0, B_k\}$. Setting $\rho_B$ equal to the appropriate global recycling constant yields exactly the structure required by (R1). □

**Proposition 6.2** (Functoriality). *The assignment*

$$(A, H) \longmapsto \mathcal{N}(A, H), \quad r \longmapsto \mathcal{N}(r)$$

*defines a functor $\mathcal{N} : \mathbf{FGraphRet} \to \mathbf{CRN}$. Moreover, by Lemma 6.1 and the description of $\mathcal{N}(r)$, the image of $\mathcal{N}$ lies in the full subcategory of CRNs satisfying* (W1)–(W6) *and* (R1).

**Lemma 6.3** (Functoriality at BP Fixed Points). *Let*

$$r : (A, H) \longrightarrow (A', H')$$

*be a morphism in $\mathbf{FGraphRet}$, and let*

$$\Gamma = \mathcal{N}(A, H), \quad \Gamma' = \mathcal{N}(A', H')$$

*be the associated CRNs. Then:*

1. *$\Gamma$ and $\Gamma'$ satisfy* (W1)–(W6) *and* (R1)*, and hence their positive steady states implement BP on $(A, H)$ and $(A', H')$, respectively, by Theorem 5.4.*

2. *The CRN morphism*

$$\mathcal{N}(r) = (\varphi_0, \varphi_1, \psi) : \Gamma \to \Gamma'$$

*deletes exactly the bundles corresponding to sites/regions removed by $r$, leaves all recycling rates $\rho_B$ for surviving bundles unchanged, and updates sum-bundle production coefficients by*

$$\lambda_{j \to n}(\mathbf{k}^j) = c_{j \to n} \psi_j(\mathbf{k}^j) \longmapsto \lambda'_{j \to n}(\mathbf{k}^j) = c_{j \to n} \psi'_j(\mathbf{k}^j)$$

*where $\psi_j \mapsto \psi'_j$ is the factor-table update induced by $r$ in the sense of Sergeant–Perthuis–Boitel.*

3. *If $\pi$ is a BP fixed point on $(A, H)$, then its restriction $\pi'$ to the surviving variables of $(A', H')$ is a BP fixed point there, and any positive steady states $x^*$ of $\Gamma$ and $x'^*$ of $\Gamma'$ whose bundle-normalized concentrations correspond to $\pi$ and $\pi'$ are related by $\mathcal{N}(r)$ on the surviving bundles.*

## 7. Simulations and Experiments

We benchmark the computational impact of SP–B retractions on CRNs obtained via the Napp–Adams compilation. For each factor-graph instance $\mathcal{F}$, we compare *compile-first* (compile $\mathcal{F}$ into $\mathcal{N}(\mathcal{F})$) against *reduce-then-compile* (apply the SP–B retraction to obtain $r(\mathcal{F})$, then compile $\mathcal{N}(r(\mathcal{F}))$). We report reductions in factor-graph size, compiled CRN

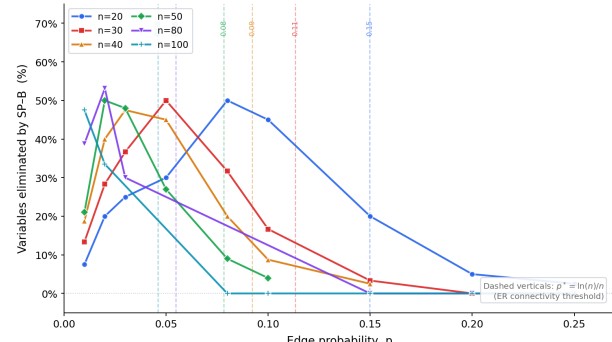

*Figure 3.* Variable reduction on $G(n, p)$ as a function of edge probability $p$. Dashed verticals mark $p^* = \ln(n)/n$. Reduction peaks below the connectivity threshold, where graphs contain small loopy cores with retractable tendrils, and falls to zero once the irreducible core dominates.

size, wall-clock simulation time for integrating the induced mass-action ODEs, and marginal agreement on the surviving variables.

Unless otherwise stated, the experiments in the main text use binary state spaces, $|E_i| = 2$, in order to isolate the effect of graph topology on reduction. This is the relevant structural variable for SP–B retractions: reducibility depends on the poset structure of the factor graph, not on the cardinalities of the variable state spaces. Larger state spaces increase the absolute number of compiled species and reactions, but do not change which variables or factors are removed by the retraction. Appendix B reports cardinality sweeps and mixed-cardinality examples confirming this behavior.

**Tree-structured factor graphs (exact BP regime).** We generate binary-tree factor graphs with depths $d \in \{3, 4, 5, 6\}$, giving $|I| \in \{7, 15, 31, 63\}$ variables. On trees, BP computes exact marginals and SP–B reductions retract away essentially all non-core acyclic structure, collapsing each instance to a trivial core. Median variable reduction is $\approx 95\%$ and median species reduction is $\approx 97.5\%$ (Table 1), yielding dramatic simulation speedups (median $\sim 885\times$). Trees are faster than chains in our benchmarks (chains: $\sim 270\times$) not because they reduce more, but because unreduced tree instances are substantially more expensive: compiled CRN size grows more aggressively with depth for trees than for chains, so the speedup ratio is larger.

**Loopy BP: loopy cores with reducible tendrils.** We generate loopy-core instances with core size $c \in \{3, 4, 5\}$ and attached tendrils of length $t \in \{1, 3, 5, 10\}$. SP–B reductions remove the acyclic tendrils while preserving the irreducible loopy core, yielding partial but substantial compression (median $\approx 79\%$ variable and $\approx 73\%$ species reduction) and corresponding speedups (median $\sim 22\times$) (Table 1).

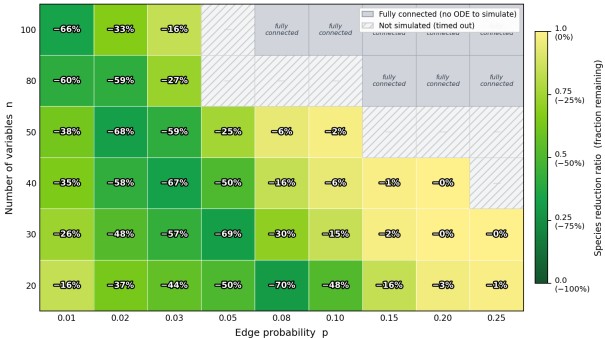

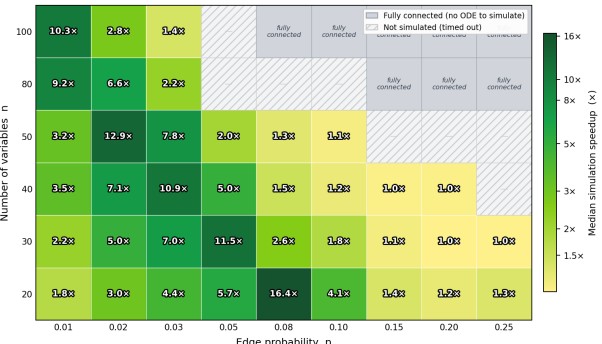

*Figure 4.* Median CRN species reduction on $G(n, p)$ instances. Species reduction is concentrated in the sparse/subcritical wedge and vanishes in the dense regime, where the irreducible loopy core dominates.

*Figure 5.* Median simulation speedup on $G(n, p)$ instances. Speedup is largest in the sparse/subcritical regime where reductions remove many species.

**Erdős–Rényi random graphs.** To test whether the method depends on planted reducible structure, we also benchmark Erdős–Rényi graphs $G(n, p)$ with $n \in [20, 100]$ and edge probabilities spanning sparse, near-threshold, and dense regimes. The relevant structural scale is the connectivity threshold $p^* = \ln(n)/n$. Below and near this threshold, graphs contain many tree-like components and tendrils attached to small loopy cores; above it, the irreducible core occupies an increasing fraction of the graph.

Figure 3 shows that variable reduction follows this core/tendril transition. Reduction is small near $p = 0$, rises in the sparse regime, peaks below $p^*$, and then falls to zero as the graph becomes dense. Species reduction shows the same pattern (Figure 4), exceeding $60\%$ in representative subcritical regimes and reaching about $69\%$ in the best median cells. Median simulation speedups peak around $16\times$ (Figure 5), with larger individual gains, but runtime is more variable because it also depends on ODE stiffness and solver behavior.

The ER benchmark therefore supports the same interpretation as the structured families: SP–B reductions remove graph regions outside the irreducible loopy core. In dense ER graphs, where the core dominates, the method correctly halts with little or no reduction rather than altering the inferential structure.

All wall-clock ER timings were measured on Auburn University's Easley high-performance computing cluster. Each job used one node, one task, one CPU per task, 8GB memory, a 12-hour wall-time limit; each array task evaluated one $(n, p)$ cell with 10 random seeds.

**Correctness diagnostics.** BP converged in all 440 ER runs on both the original and the reduced networks. We additionally track three marginal-agreement diagnostics: BP(original) vs. BP(reduced) on surviving variables (the SP–B preservation test), CRN-derived marginals vs. BP

marginals on each network (end-to-end validation of the compiled chemical implementation), and CRN(original) vs. CRN(reduced) on surviving species.

For tree instances, BP is exact and the BP(orig) vs. BP(reduced) comparison is a stringent correctness test; observed differences are dominated by numerical precision . For loopy and ER instances, we interpret the comparison as a core fixed-point preservation diagnostic. The median BP(original) vs. BP(reduced) discrepancy on ER instances is $4 \times 10^{-8}$. A small fraction of near-critical runs ($\sim 1\%$ above $0.1$) show larger discrepancies; these correspond to graphs near the connectivity threshold where loopy BP can have multiple fixed points and the original and reduced systems converge to different basins. These gains reflect computational savings from retraction rather than changes in the inferred beliefs, with the caveat that loopy BP non-uniqueness can select different fixed points across the two systems in the near-critical regime.

## Future Work

A natural next step is physical realization via DNA strand displacement, which by the universality result of (Soloveichik et al., 2010) can implement any mass-action CRN. Our reduction pipeline therefore provides a direct path to smaller, more experimentally feasible DNA implementations of BP inference: reduce the factor graph, compile via Napp-Adams, then implement the reduced CRN in DNA. The species counts reported here suggest this could bring moderately complex inference tasks within reach of current DNA nanotechnology. A longer-horizon question is whether gene regulatory networks that perform approximate inference can be cast in Napp-Adams form, which would make them amenable to the same reduction pipeline; this depends on whether their ODE structure can be shown to satisfy conditions (W1)–(W6), and we leave this as an open problem.

## Impact Statement

We are not aware of any immediate negative societal impacts arising from this work. Methods that make molecular computation more physically realizable could contribute to the design of programmable cells and synthetic biological systems. The ethical implications of such downstream applications would need to be assessed as the technology matures.

## Acknowledgments

This material is based upon research supported by the Chateaubriand Fellowship of the Office for Science Technology of the Embassy of France in the United States as well as by NSF Grant DMS-2246127-PSC and DOE Grant DE-SC0025649

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

## Reduction of CRNs

A common class of exact reduction methods exploits stoichiometric structure to eliminate subnetworks while preserving steady-state behavior of the unreduced portion. In particular, Hirono et al. (Hirono et al., 2021) identify output-complete subnetworks $\gamma$ and compute an *influence index* $\lambda(\gamma)$, an integer determined by its topology, $\chi(\gamma) = |V_\gamma| - |E_\gamma|$ together with stoichiometric cycle and conservation subspaces. When $\lambda(\gamma) = 0$, they prove a localization principle: steady-state sensitivities outside $\gamma$ are unaffected by perturbations internal to $\gamma$, which justifies eliminating $\gamma$ via a Schur complement reduction on the stoichiometry. Related reduction strategies apply Schur complements to weighted graph Laplacians derived from CRNs. For complex-balanced mass-action systems, Rao et al. (Rao et al., 2013) use the weighted Laplacian on the graph of complexes to delete complexes while retaining equilibrium structure, and later work (Gasparyan et al., 2025) proposes a Schur complement on *species–reaction graphs* with an emphasis on approximation error in dynamical trajectories. Broader families of model simplification and reduction such as conservation analysis, decomposition, timescale separation, and sensitivity-based methods are surveyed in (Snowden et al., 2017).

However, these methods do not apply directly to CRNs arising from hidden Markov models, graphical models, and factor graphs (Wiuf et al., 2023; Napp & Adams, 2013; Colliaux et al., 2017). In these networks, key reactions are catalyzed, and the stoichiometric matrix alone fails to capture the underlying message-passing structure. We therefore exploit reduction techniques developed for graphical models that preserve collections of distributions corresponding to fixed points of the belief propagation algorithm (Sergeant-Perthuis & Boitel, 2025; Yedidia et al., 2000). To this end, we propose a dictionary that maps CRNs to graphical models in a structure-preserving manner.

## A. Conceptual Overview and Scope

This section makes explicit the dictionary used throughout Sections 4–6. Table 2 summarizes the correspondence between factor-graph concepts and their chemical realizations in a Napp–Adams compiled CRN; conditions (W1)–(W6) formalize the right column, and (R1) ensures the resulting steady states are precisely the BP fixed-point equations after bundle normalization. Table 3 clarifies which classes of CRNs fall within and outside this framework.

*Table 2.* Factor-graph/CRN correspondence in a Napp–Adams compiled CRN.

| Concept | Factor graph | CRN |
|---|---|---|
| Variable | Variable node, state space $E_i$ | Equivalence class of product bundles (Definition 5.1) |
| Factor | Factor node $j$, potential $\psi_j$ | Factor class of sum bundles (W6) |
| Message | $P^{(i \to j)}$, $S^{(j \to i)}$ | Bundle-normalized concentration vector (Theorem 5.4) |
| Factor update | Summation over neighboring variables | Catalyzed production into sum bundle (W4, W6) |
| Product update | Product of incoming messages | Catalyzed production into product bundle (W5) |
| Retraction | SP–B deletion and table update | Bundle deletion plus rate-table update (Proposition 6.2) |

*Table 3.* Scope of the recognition and reduction framework.

| CRN class | W1–W6 | R1 | Pipeline | Interpretation |
|---|---|---|---|---|
| Napp–Adams compiled BP CRNs (Napp & Adams, 2013) | ✓ | ✓ | ✓ | BP present by construction. |
| CRNs after SP–B reduction | ✓ | ✓ | ✓ | Closed under reduction by Proposition 6.2. Reduced CRN remains a compiled BP implementation. |
| Baum–Welch HMM CRNs (Wiuf et al., 2023) | outside | outside | outside | Futile-cycle architecture implements EM/Baum–Welch learning rather than Napp–Adams sum-product BP. |
| General biochemical CRNs | outside | outside | outside | Not expected to have message bundles, factor-table rate structure, or BP-normalizing recycling. |

Entries marked "outside" indicate only that the relevant semantics differ from the BP message-bundle structure captured by (W1)–(W6) and (R1), not that such CRNs are unreducible by other means.

## B. State-Cardinality Sweeps

The main experiments use binary variables in order to isolate the role of graph topology in SP–B reduction. Here we report additional experiments showing that the same reduction behavior persists for larger state spaces. The reason is structural: SP–B reducibility is determined by the factor-graph poset, not by the cardinality of the variable state spaces. Increasing the alphabet size changes the absolute number of compiled CRN species and reactions, but it does not change which variables or factors are linear or colinear points.

We tested cardinalities $K \in \{2, 3, 4, 5\}$ on chains, depth-4 trees, and loopy-core graphs with tendrils. We also tested a mixed-cardinality chain. For each instance, we apply the same reduce-then-compile pipeline as in Section 7, and we report variable reduction, species reduction, and maximum BP marginal discrepancy on the surviving variables.

*Table 4.* State-cardinality sweeps. Reduction percentages remain stable across cardinalities because SP–B reducibility depends on graph topology rather than alphabet size. Here $K$ denotes the common state-space cardinality of each variable, except in the mixed-cardinality row.

| Family | Cardinalities | Variables | Species | Max BP discrepancy |
|---|---|---|---|---|
| Chain | $K = 2, 3, 4, 5$ | $10 \to 1$ | $(150, 200, 250, 300) \to (9, 12, 15, 18)$ | $7.93 \times 10^{-10}$ to $5.55 \times 10^{-17}$ |
| Tree | $K = 2, 3, 4, 5$ | $15 \to 1$ | 96.6% reduction | near numerical precision |
| Loopy+tendril | $K = 2, 3, 4, 5$ | $16 \to 4$ | 68.2% reduction | near numerical precision for $K = 4, 5$ |
| Mixed chain | $[2, 3, 4, 5, 4, 3, 2]$ | $7 \to 1$ | 94.0% reduction | $5.79 \times 10^{-13}$ |

For the 10-variable chains, the compiled CRN sizes grow linearly with $K$ in this family: the unreduced networks have $150, 200, 250,$ and $300$ species for $K = 2, 3, 4, 5$, respectively, while the reduced networks have $9, 12, 15,$ and $18$ species. Thus the variable reduction is $90.0\%$ and the species reduction is $94.0\%$ in every case. The BP discrepancy on the surviving variable remains negligible, decreasing from $7.93 \times 10^{-10}$ for $K = 2$ to $5.55 \times 10^{-17}$ for $K = 5$.

The tree and loopy-core families show the same qualitative behavior. Depth-4 trees reduce from 15 variables to 1 variable, with $96.6\%$ species reduction across all tested cardinalities. Loopy-core graphs with tendrils reduce from 16 variables to 4 variables, with $68.2\%$ species reduction across all tested cardinalities. In the loopy case, the reduction removes the tendrils while preserving the irreducible core, so increasing the state-space cardinality changes the size of the compiled CRN but not the graph-theoretic core.

Finally, the mixed-cardinality chain $[2, 3, 4, 5, 4, 3, 2]$ reduces from 7 variables to 1 variable, with $94.0\%$ species reduction and BP discrepancy $5.79 \times 10^{-13}$. This confirms that the reduction does not require all variables to have the same alphabet size. The state spaces affect the sizes of the species bundles and factor tables, while reducibility is controlled by the underlying factor-graph topology.

## C. Topological Background

**Definition C.1** (Alexandrov topology). Let $(X, \leq)$ be a poset. The *Alexandrov topology* on $X$ is the topology whose open sets are the *up-sets*:

$$U \subseteq X \text{ is open} \quad \Longleftrightarrow \quad (\forall x \in U)(\forall y \in X)\,(x \leq y \Rightarrow y \in U).$$

Equivalently, the closed sets are exactly the *down-sets*:

$$C \subseteq X \text{ is closed} \quad \Longleftrightarrow \quad (\forall x \in C)(\forall y \in X)\,(y \leq x \Rightarrow y \in C).$$

We write $X_{\mathcal{A}}$ for the topological space obtained from a poset $\mathcal{A}$ equipped with its Alexandrov topology.

*Remark* C.2. A map between Alexandrov spaces is continuous precisely when it is order-preserving on the underlying posets.

### Homotopy for finite posets

We adopt the notion of homotopy for finite posets used in (Sergeant-Perthuis & Boitel, 2025). Informally, one views a finite poset $\mathcal{A}$ as an Alexandrov space $X_{\mathcal{A}}$ and uses continuous (equivalently, order-preserving) maps to the target poset $X_{\mathcal{A}} \times [0, 1] \to X_{\mathcal{B}}$.

**Definition C.3** (Deformation retracts of posets). Let $\mathcal{B} \subseteq \mathcal{A}$ be a subposet and let $i : \mathcal{B} \hookrightarrow \mathcal{A}$ be the inclusion. A *deformation retract* of $\mathcal{A}$ onto $\mathcal{B}$ consists of an order-preserving map $r : \mathcal{A} \to \mathcal{B}$ such that

$$r \circ i = \mathrm{id}_{\mathcal{B}} \qquad \text{and} \qquad i \circ r \simeq \mathrm{id}_{\mathcal{A}},$$

where $\simeq$ denotes homotopy in the sense of (Sergeant-Perthuis & Boitel, 2025). It is a *strong deformation retract* if, moreover, the homotopy from $i \circ r$ to $\mathrm{id}_{\mathcal{A}}$ fixes $\mathcal{B}$ pointwise.

*Remark* C.4. Two finite posets $\mathcal{A}$ and $\mathcal{B}$ are *homotopy equivalent* if there exist order-preserving maps $\phi : \mathcal{A} \to \mathcal{B}$ and $\psi : \mathcal{B} \to \mathcal{A}$ with $\psi \circ \phi \simeq \mathrm{id}_{\mathcal{A}}$ and $\phi \circ \psi \simeq \mathrm{id}_{\mathcal{B}}$ (in the same sense as above).

### Linear/colinear points and cores

The reductions we use are based on deleting elements of a finite poset without changing its homotopy type. The relevant notions are the following.

**Definition C.5** (Linear and colinear points). Let $\mathcal{A}$ be a finite poset and let $a \in \mathcal{A}$.

**(Linear).** We say $a$ is *linear* if there exists an element $a^{\uparrow} \in \mathcal{A}$ with $a < a^{\uparrow}$ such that every element above $a$ is also above $a^{\uparrow}$, i.e.

$$(\forall b \in \mathcal{A})\,\left(b \geq a \Rightarrow b \geq a^{\uparrow}\right). \tag{L}$$

**(Colinear).** Dually, we say $a$ is *colinear* if there exists $a^{\downarrow} \in \mathcal{A}$ with $a^{\downarrow} < a$ such that every element below $a$ is also below $a^{\downarrow}$, i.e.

$$(\forall b \in \mathcal{A})\,\left(b \leq a \Rightarrow b \leq a^{\downarrow}\right). \tag{coL}$$

*Remark* C.6. Condition (L) says that $a^{\uparrow}$ is a *least* element among those strictly above $a$ (up to comparability): once you pass above $a$, you have automatically passed above $a^{\uparrow}$. The colinear condition is the order-dual statement. These are exactly the points that can be removed by an elementary up- or down-retraction without changing homotopy type (Stong, 1966; Sergeant-Perthuis & Boitel, 2025).

**Definition C.7** (Core of a finite poset (Stong, 1966)). The *core* of a finite poset $\mathcal{A}$ is a subposet $\mathcal{B} \subseteq \mathcal{A}$ such that:

1. $\mathcal{B}$ has no linear or colinear points (computed within $\mathcal{B}$), and

2. $\mathcal{B}$ is a strong deformation retract of $\mathcal{A}$ (Definition C.3).

We denote a core of $\mathcal{A}$ by $co\mathcal{A}$.

**Proposition C.8.** *Every finite poset $\mathcal{A}$ admits a core $co\mathcal{A}$. Moreover, one can obtain a strong deformation retract $\mathcal{A} \to co\mathcal{A}$ by iteratively applying up- and down-retractions that remove linear and colinear points. Finally, two finite posets are homotopy equivalent if and only if their cores are isomorphic.*

*Proof.* This is Theorem 2 of (Stong, 1966). $\square$

# D. Proofs of propositions, lemmas, and theorems

*Proof of Theorem 4.7.* The factor classes $J$ from (W6) and variable classes $[P]$ from Definition 5.1 immediately provide the node sets. By Lemma 5.3, if $P \sim_v P'$ then $|E_P| = |E_{P'}|$, so the state space $E_{[P]}$ is well-defined up to the canonical bijections induced by (W5) and (W6).

We wish to show that the incidence relation is well defined, which amounts to showing that for each factor class $J$ and distinct $Q, Q' \in V_J$, the variable classes $[Q]$ and $[Q']$ are distinct.

Suppose for contradiction that $Q \sim_v Q'$ for some $Q \neq Q'$ in $V_J$. Then there exists a chain $Q = P_0 \sim P_1 \sim \cdots \sim P_m = Q'$ of direct cross-links. Consider the first link: $Q \sim P_1$ means there exists a sum bundle $B$ with either $\text{Tgt}(B) = Q$ and $B \in \text{Cat}(P_1)$, or $\text{Tgt}(B) = P_1$ and $B \in \text{Cat}(Q)$.

In the first case, let $J'$ be the factor class containing $B$, so $B = B_{J' \to Q}$ and hence $Q \in V_{J'}$. Since $B \in \text{Cat}(P_1)$ and $P_1$ is product-like, condition (W5) tells us that $B$ catalyzes production of $P_1$. By the structure of (W6), the sum bundle $B$ targets $Q$ and has $\text{Cat}(B) = V_{J'} \setminus \{Q\}$. For $B$ to appear in $\text{Cat}(P_1)$, the product bundle $P_1$ must be receiving messages from the same variable as $Q$ but through a different factor than $J'$. In particular, $P_1 \notin V_{J'}$.

The second case is symmetric: if $\text{Tgt}(B) = P_1$ and $B \in \text{Cat}(Q)$, then $P_1$ is the target of some sum bundle while $Q$ is catalyzed by that bundle, again placing $Q$ and $P_1$ as product bundles for the same variable going to different factors.

In either case, each cross-link $P_i \sim P_{i+1}$ in the chain connects product bundles that represent the same variable but are incident to different factor classes. The key observation is that the relation $\sim$ *cannot* connect two product bundles that are both in the same $V_J$: if $Q, Q' \in V_J$ with $Q \neq Q'$, then $Q$ and $Q'$ are targets of distinct sum bundles in $J$, namely $B_{J \to Q}$ and $B_{J \to Q'}$. By (W6.1), $\text{Cat}(B_{J \to Q}) = V_J \setminus \{Q\}$ and $\text{Cat}(B_{J \to Q'}) = V_J \setminus \{Q'\}$.

For $Q \sim Q'$ directly, we would need a sum bundle $B$ with $\text{Tgt}(B) = Q$ and $B \in \text{Cat}(Q')$ (or vice versa). If $B \in J$, then $B = B_{J \to Q}$, and $B \in \text{Cat}(Q')$ would require $Q'$ to be a product bundle catalyzed by this sum bundle. But $\text{Cat}(B_{J \to Q}) = V_J \setminus \{Q\}$ consists of product bundles in $V_J$, not the catalytic neighbors of those bundles. The set $\text{Cat}(Q')$ consists of sum bundles that catalyze $Q'$, which by the Napp–Adams structure are sum bundles targeting the same variable as $Q'$ from other factors. Since $B_{J \to Q}$ targets $Q$ (a different element of $V_J$, hence a different variable), we have $B_{J \to Q} \notin \text{Cat}(Q')$.

If instead $B \notin J$, say $B \in J'$ for some other factor class, then $\text{Tgt}(B) = Q$ implies $Q \in V_{J'}$. For $B \in \text{Cat}(Q')$, the sum bundle $B$ must be one of the catalytic inputs to $Q'$. These are sum bundles for the variable corresponding to $Q'$, coming from factors other than the one $Q'$ points to. Since $B$ targets $Q$ and $Q \in V_J$, the variable of $B$ is the variable of $Q$. For $B \in \text{Cat}(Q')$, the variable of $Q'$ must equal the variable of $Q$. But $Q$ and $Q'$ are distinct elements of $V_J$, which by (W6.1) correspond to distinct targets of sum bundles in $J$—hence to distinct variables incident to factor $J$. This is a contradiction.

Therefore $Q \not\sim Q'$ for distinct $Q, Q' \in V_J$, and since $\sim_v$ is generated by $\sim$, we have $Q \not\sim_v Q'$. This shows $[Q] \neq [Q']$, and moreover $[Q] \cap V_J = \{Q\}$ (any other element of $[Q]$ in $V_J$ would have to equal $Q$ by the above).

Consequently, the map $Q \mapsto [Q]$ is a bijection from $V_J$ to the set of variable classes incident to $J$. The incidence relation is well-defined, and the factor table $\psi_J : \prod_{Q \in V_J} E_Q \to (0, \infty)$ from (W6.3) can be viewed as a function on $\prod_{[Q]:Q \in V_J} E_{[Q]}$ via the canonical identifications $E_Q \cong E_{[Q]}$.

Finally, uniqueness of $\psi_J$ up to positive scalar follows from (W6.3): the coefficient arrays $C_{B_{J \to Q}}$ are determined by $\Gamma$, and any two tables $\psi_J, \psi'_J$ satisfying (W6.3) must have constant ratio $\psi'_J / \psi_J = \lambda$ for some $\lambda > 0$, with the scalars $\kappa_{J \to Q}$ adjusting accordingly. $\square$

*Proof of Lemma 5.2.* The reflexive-symmetric-transitive closure of any binary relation is an equivalence relation. □

*Proof of Lemma 5.3.* It suffices to show $P \sim P'$ implies $|E_P| = |E_{P'}|$; the general case follows by induction along $\sim$-chains.

Suppose $P \sim P'$ via sum bundle $B$ with $\mathrm{Tgt}(B) = P$ and $B \in \mathrm{Cat}(P')$. Let $J$ be the factor class containing $B$, so $B = B_{J \to P}$ and $P \in V_J$.

- By (W6.2), $\iota_{J \to P} : E_P \xrightarrow{\cong} E_B$, so $|E_P| = |E_B|$.

- Since $B \in \mathrm{Cat}(P')$ and $P'$ is product-like, (W5.2) gives a bijection $\theta_{P',B} : E_{P'} \to E_B$, so $|E_{P'}| = |E_B|$.

Hence $|E_P| = |E_B| = |E_{P'}|$. □

*Proof of Theorem 5.4.* Fix a factor class $(J, V_J)$ and a product bundle $Q \in V_J$. Consider the associated sum bundle $B := B_{J \to Q} \in J$ from (W6). By conditions (W1)–(W3), every reaction with nonzero net stoichiometry in $B$ has internal net stoichiometry of the form $B_0 \leftrightarrow B_k$ for a unique $k \in E_B$, and no other bundle has nonzero net stoichiometry in that reaction.

For each $k \in E_B$, let $\frac{d}{dt}[B_k]^+$ denote the total positive contribution to $\frac{d}{dt}[B_k]$ coming from reactions whose internal net stoichiometry is $B_0 \to B_k$. By (W4), we may expand:

$$\frac{d}{dt}[B_k]^+ = [B_0] \sum_{a \in \mathcal{A}_B} C_B(k; a) \prod_{Q' \in \mathrm{Cat}(B)} [X_{Q', a(Q')}].$$

By the missing-one-neighbor property of (W6.1), we have $\mathrm{Cat}(B) = V_J \setminus \{Q\}$. Hence each assignment $j \in J_B = \prod_{Q' \in V_J \setminus \{Q\}} E_{Q'}$ is equivalently a choice of states $(x_{Q'})_{Q' \neq Q}$ for all other product bundles in $V_J$.

Now apply the clamped-table property (W6.3): there exists a strictly positive table $\psi_J : \prod_{Q' \in V_J} E_{Q'} \to (0, \infty)$ and a scalar $\kappa_{J \to Q} > 0$ such that for every $k \in E_Q$ (identified with $E_B$ via the bijection $\iota_{J \to Q}$) and every assignment $a \in \prod_{Q' \in V_J \setminus \{Q\}} E_{Q'}$:

$$C_B(\iota_{J \to Q}(k); a) = \kappa_{J \to Q} \cdot \psi_J\big(x_Q = k, \, x_{Q'} = a(Q') \text{ for } Q' \neq Q\big).$$

Substituting this identity into the expansion for $\frac{d}{dt}[B_k]^+$:

$$\frac{d}{dt}[B_k]^+ = [B_0]\kappa_{J \to Q} \sum_{a \in \prod_{Q' \in V_J \setminus \{Q\}} E_{Q'}} \psi_J\big(x_Q = k, \, x_{Q'} = a(Q')\big) \prod_{Q' \in V_J \setminus \{Q\}} [Q'_{a(Q')}].$$

Next, we identify the negative contribution to the ODE. By (R1), the only internal reactions with net stoichiometry $B_k \to B_0$ are the recycling reactions $B_k \xrightarrow{\rho_B} B_0$, with no catalysts and a bundlewise constant rate $\rho_B = \rho_{B_{J \to Q}}$. Hence the mass-action dynamics for coordinate $B_k$ are:

$$\frac{d}{dt}[B_k] = \frac{d}{dt}[B_k]^+ - \rho_B[B_k].$$

At a positive steady state, $\frac{d}{dt}[B_k] = 0$. Rearranging:

$$\frac{\rho_B}{[B_0]}[B_k] = \kappa_{J \to Q} \sum_{a \in \prod_{Q' \in V_J \setminus \{Q\}} E_{Q'}} \psi_J\big(x_Q = k, \, x_{Q'} = a(Q')\big) \prod_{Q' \in V_J \setminus \{Q\}} [Q'_{a(Q')}]. \tag{3}$$

This establishes equation (1).

Let $P$ be a product bundle. By (W5), $P$ is product-like: for each $k \in E_P$ there is a unique assignment $a^{(k)} \in \mathcal{A}_P$ supporting production of $P_k$, and along each catalytic neighbor the induced map on state labels is a bijection.

By the structure of the Napp–Adams construction (which our recognition conditions capture), the producing reactions for $P_k$ have the schematic form:

$$P_0 + \sum_{B \in \text{Cat}(P)} B_k \longrightarrow P_k + \sum_{B \in \text{Cat}(P)} B_k,$$

where one coordinate labeled $k$ from each incident sum bundle catalyzes production of the coordinate labeled $k$ in $P$. (Here we use the bijections from (W5) to identify state labels across bundles.)

Under (R1), every such producing reaction has rate constant $\kappa_P$ independent of $k$. Thus the total production rate into $P_k$ is:

$$\frac{d}{dt}[P_k]^+ = \kappa_P[P_0] \prod_{B \in \text{Cat}(P)} [B_k].$$

Here $\text{Cat}(P)$ denotes the set of sum bundles incident to $P$—equivalently, for each factor class $J$ containing $P$ in its variable set $V_J$, we include the sum bundle $B_{J \to P}$.

Again by (R1), the only internal reactions with net stoichiometry $P_k \to P_0$ are the recycling reactions $P_k \xrightarrow{\rho_P} P_0$. Hence:

$$\frac{d}{dt}[P_k] = \kappa_P[P_0] \prod_{B \in \text{Cat}(P)} [B_k] - \rho_P[P_k].$$

At a positive steady state:

$$\frac{\rho_P}{\kappa_P[P_0]}[P_k] = \prod_{B \in \text{Cat}(P)} [B_k]. \tag{4}$$

This establishes equation (2).

We now show that equations (3) and (4) are precisely the sum-product BP fixed-point equations on the reconstructed factor graph $\text{FG}(\Gamma)$.

Recall from Theorem 4.7 that $\text{FG}(\Gamma)$ has:

- Factor nodes corresponding to factor classes $J$;
- Variable nodes corresponding to variable classes $[P]$ (equivalence classes of product bundles under $\sim_v$);
- An edge between factor $J$ and variable $[P]$ whenever $[P] \cap V_J \neq \emptyset$;
- Factor table $\psi_J : \prod_{Q \in V_J} E_Q \to (0, \infty)$ for each factor $J$.

In the standard BP notation on this factor graph:

- $S_k^{(j \to n)}$ denotes the $k$-th component of the sum message from factor $j$ to variable $n$;
- $P_k^{(n \to j)}$ denotes the $k$-th component of the product message from variable $n$ to factor $j$.

The BP sum-product fixed-point equations are (cf. (Napp & Adams, 2013)):

$$S_k^{(j \to n)} \propto \sum_{\mathbf{k}^j : k_n^j = k} \psi_j(\mathbf{k}^j) \prod_{n' \in \text{ne}(j) \backslash n} P_{k_{n'}^j}^{(n' \to j)}, \tag{5}$$

$$P_k^{(n \to j)} \propto \prod_{j' \in \text{ne}(n) \backslash j} S_k^{(j' \to n)}. \tag{6}$$

We establish the correspondence by setting:

$$S_k^{(J \to [Q])} := [B_{J \to Q, k}],$$
$$P_k^{([Q] \to J)} := [P_{Q, k}],$$

where $Q$ is the unique representative of $[Q]$ in $V_J$.

**Sum messages:** In our CRN, the sum bundle $B_{J \to Q}$ corresponds to the message from factor $J$ to variable $[Q]$. Equation (3) states:

$$\frac{\rho_B}{[B_0]}[B_k] = \kappa_{J \to Q} \sum_a \psi_J(x_Q = k, x_{Q'} = a(Q')) \prod_{Q' \in V_J \setminus \{Q\}} [Q'_{a(Q')}].$$

Since product bundles $Q' \in V_J \setminus \{Q\}$ correspond to the other variables incident to factor $J$, and $[Q'_{a(Q')}]$ is the concentration of the product-message species, we have:

$$[B_k] \propto \sum_{\mathbf{k}^J : k_Q^J = k} \psi_J(\mathbf{k}^J) \prod_{Q' \in V_J \setminus \{Q\}} [Q'_{k_{Q'}^J}],$$

which matches equation (5) under our identification.

**Product messages:** For a product bundle $P$ corresponding to variable $[P]$ sending to factor $J$, equation (4) states:

$$\frac{\rho_P}{\kappa_P [P_0]}[P_k] = \prod_{B \in \mathrm{Cat}(P)} [B_k].$$

The catalytic neighbors $\mathrm{Cat}(P)$ are exactly the sum bundles from factors $J' \neq J$ incident to variable $[P]$. Hence:

$$[P_k] \propto \prod_{J' \in \mathrm{ne}([P]) \setminus J} [B_{J' \to P, k}],$$

which matches equation (6) under our identification.

**Normalization:** The proportionality constants $\rho_B/[B_0]$ and $\rho_P/(\kappa_P[P_0])$ serve as the per-message normalization factors. In standard BP, messages are defined only up to positive rescaling, and these normalization factors are precisely what allow the concentrations to be interpreted as (unnormalized) message components.

Since the sum of concentrations within each bundle is conserved (a consequence of the stoichiometric structure), normalization is automatically maintained by the reaction dynamics.

This completes the identification of CRN steady states with BP fixed points. $\qquad\square$

**Definition D.1** (Napp Compilation). Given a factor graph $(A, H)$ with factor tables $\psi_j \propto e^{-H_j}$, the *Napp compilation* is the mass-action CRN

$$\mathcal{N}(A, H)$$

constructed as follows:

1. For every directed edge $j \to n$ in the factor graph, introduce a sum bundle $\mathsf{S}^{j \to n}$ and a product bundle $\mathsf{P}^{n \to j}$. Each bundle $B$ consists of a zero species $B_0$ and one coordinate species $B_k$ for every symbol $k$ of the underlying variable.

2. For each bundle $B$, add recycling reactions:

$$B_k \xrightarrow{\kappa_r} B_0, \quad k \geq 1,$$

   with a single global recycling rate $\kappa_r > 0$.

3. For each edge $j \to n$ and each assignment $\mathbf{k}^j \in K^j$ with $(\mathbf{k}^j)_n = k$, add a sum-bundle production reaction:

$$\mathsf{S}_0^{j \to n} + \sum_{n' \in \mathrm{ne}(j) \setminus n} \mathsf{P}_{(\mathbf{k}^j)_{n'}}^{n' \to j} \xrightarrow{\psi_j(\mathbf{k}^j)} \mathsf{S}_k^{j \to n} + \sum_{n' \in \mathrm{ne}(j) \setminus n} \mathsf{P}_{(\mathbf{k}^j)_{n'}}^{n' \to j}.$$

4. For each variable $n$ and each state $k$, add product-bundle reactions:

$$\mathsf{P}_0^{n \to j} + \sum_{j' \in \mathrm{ne}(n) \setminus j} \mathsf{S}_k^{j' \to n} \xrightarrow{\kappa_{\mathrm{prod}}} \mathsf{P}_k^{n \to j} + \sum_{j' \in \mathrm{ne}(n) \setminus j} \mathsf{S}_k^{j' \to n},$$

   with a single global production rate $\kappa_{\mathrm{prod}} > 0$.

We write $\mathcal{N}(A, H)$ (or simply $\mathcal{N}(\mathrm{FG})$) for this CRN and refer to the map $\mathcal{N} : \mathbf{FGraph} \to \mathbf{CRN}$ as the Napp compilation.

**Corollary D.2** (BP $\leftrightarrow$ CRN). *Let $\Gamma = (X, R, \kappa)$ be a mass–action CRN satisfying* (W1)–(W6) *and* (R1)*, and let* $\mathrm{FG}(\Gamma)$ *be the factor graph with factor tables* $\{\psi_J\}$ *reconstructed by Theorem 4.7. Let*

$$\Gamma^{\mathrm{N}} := \mathcal{N}\big(\mathrm{FG}(\Gamma)\big)$$

*be the BP–CRN obtained by applying the Napp–Adams compilation to* $\mathrm{FG}(\Gamma)$*, with any fixed choice of recycling and production constants.*

*Then there exist positive scalars* $\{s_B > 0\}$*, one for each bundle $B$, and a choice of rate constants for* $\Gamma^{\mathrm{N}}$ *such that the following hold.*

1. *The species of $\Gamma$ and $\Gamma^{\mathrm{N}}$ can be put into bijection bundlewise, preserving zero versus coordinate species, the distinction between sum and product bundles, and catalytic neighbor sets, and under this bijection the change of variables*

$$[X_{B,k}]^{\mathrm{N}} = s_B\,[X_{B,k}]$$

   *conjugates the mass–action vector field of $\Gamma$ to that of $\Gamma^{\mathrm{N}}$.*

2. *In particular, positive steady states of $\Gamma$ correspond bijectively, via the same bundlewise rescaling, to positive steady states of $\Gamma^{\mathrm{N}}$.*

3. *From the main result of Napp–Adams applied to $\Gamma^{\mathrm{N}}$, any positive steady state of $\Gamma$ therefore determines a collection of BP messages (defined up to the usual per–message positive rescaling) on $\mathrm{FG}(\Gamma)$ that satisfy the sum–product fixed–point equations for the recovered factor tables $\{\psi_J\}$.*

*Proof of D.2.* Let $\mathrm{FG}(\Gamma)$ be reconstructed from $\Gamma$ under (W1)–(W6), with factor classes $(J, V_J)$ and factor tables $\psi_J$ from (W6). Consider the Napp compilation $\Gamma^N := \mathcal{N}(\mathrm{FG}(\Gamma))$, in which each directed incidence $(J, Q)$ with $Q \in V_J$ produces a sum bundle $\mathsf{S}^{J \to Q}$ and each directed incidence $(Q, J)$ produces a product bundle $\mathsf{P}^{Q \to J}$.

By the construction of $\mathrm{FG}(\Gamma)$ in Theorem 4.7, the bundles of $\Gamma$ decompose into:

1. Sum bundles $B_{J \to Q}$ indexed by pairs $(J, Q)$ with $Q \in V_J$;

2. Product bundles $P$ indexed by directed incidences $(Q, J)$.

The Napp compilation produces exactly one bundle of each type for each such directed incidence. Hence there is a canonical bundlewise bijection between the species sets of $\Gamma$ and $\Gamma^N$, preserving bundle type (sum vs. product), zero vs. coordinate species, and state labels.

Moreover, by (W4) and (W5), the reaction monomials of $\Gamma$ match the Napp monomials:

- Sum-bundle production terms are multilinear monomials with one coordinate from each catalytic neighbor;

- Product-bundle production terms are diagonal in the state label $k$.

Fix a factor class $(J, V_J)$ and $Q \in V_J$. By (W6.3), the production coefficients of the corresponding sum bundle $B_{J \to Q}$ satisfy:

$$C_{B_{J \to Q}}(k; a) = \kappa_{J \to Q} \cdot \psi_J(x_Q = k, x_{Q'} = a(Q') \text{ for } Q' \neq Q).$$

In the Napp compilation, the analogous producing reactions for $\mathsf{S}_k^{J \to Q}$ have rate constants proportional to the same clamped table entries of $\psi_J$, with a user-chosen proportionality constant. Choose these Napp proportionality constants so that, after a bundlewise rescaling

$$[X_{B,k}]^N = s_B[X_{B,k}],$$

the coefficients of every monomial in the sum-bundle production polynomials agree term-by-term between $\Gamma$ and $\Gamma^N$.

Similarly, by (R1) we may choose Napp recycling and product-production constants to match the internal $k$-uniform recycling and $k$-uniform product production in $\Gamma$ (again after the same bundlewise rescaling).

Because mass-action vector fields are polynomial and determined by the reaction monomials and their coefficients, the term-by-term agreement implies that the change of variables $[X_{B,k}]^N = s_B[X_{B,k}]$ conjugates the ODE of $\Gamma$ to that of $\Gamma^N$. Since the vector fields are conjugate under a linear (diagonal, positive) change of coordinates, positive steady states correspond bijectively under this rescaling. A point $x^* \in \mathbb{R}^M_{>0}$ is a steady state of $\Gamma$ if and only if the rescaled point $(s_B x^*_{B,k})$ is a steady state of $\Gamma^N$. Theorem 5.4 establishes that positive steady states of any CRN satisfying (W1)–(W6) and (R1) correspond to BP fixed points on the reconstructed factor graph. Applying this to both $\Gamma$ and $\Gamma^N$:

- Any positive steady state $x^*$ of $\Gamma$ corresponds to a BP fixed point on FG($\Gamma$);

- The rescaled steady state $(s_B x^*_{B,k})$ of $\Gamma^N$ corresponds to the same BP fixed point (since rescaling messages by positive constants does not change BP fixed-point status).

Thus steady states of $\Gamma$ determine BP fixed points on FG($\Gamma$), with factor tables $\{\psi_J\}$ recovered from the reaction coefficients. $\qquad\square$

### D.1. SP–B retractions in the space of factor tables

We briefly recall the SP–B update rules in the space of factor tables. Write $\psi_c \propto e^{-H_c}$.

- **Linear retraction.** The Hamiltonians on survivors are unchanged (restriction), hence $\psi'_c = \psi_c$ for all surviving $c$.

- **Colinear retraction.** Let $a$ be maximal colinear with lower neighbor $a^\downarrow$. Let $B = \mathcal{A} \setminus \{a\}$. Let $b^\uparrow$ be the unique upper cover of $a^\downarrow$ in $B$. Define
$$\widehat{H}_{a^\downarrow}(x) = \ln \sum_{z \in E_a:\, z_{a^\downarrow} = x} \exp\{-H_a(z) + H_{a^\downarrow}(x)\}.$$

Then $\widetilde{H}_c = H_c$ for $c \neq b^\uparrow$ and $\widetilde{H}_{b^\uparrow} = H_{b^\uparrow} - \widehat{H}_{a^\downarrow} \circ \pi_{b^\uparrow a^\downarrow}$. Exponentiating gives the two standard factor–table forms:

$$(\text{Unary} \to \text{higher parity}) \quad \psi'_{b^\uparrow}(\mathbf{k}) = \psi_{b^\uparrow}(\mathbf{k})\, \psi_a(k_{a^\downarrow}),$$

$$(\text{Higher parity} \to \text{unary on } a^\downarrow) \quad \psi'_{a^\downarrow}(x) = \sum_{z:\, z_{a^\downarrow} = x} \psi_a(z).$$

**Lemma D.3** (Factor version of S–P Eq. 4.29: colinear, survivor linear). *Let $a \in \mathcal{A}$ be colinear and maximal, let $B = \mathcal{A} \setminus \{a\}$, and let $b = a^\downarrow \in B$ be* linear *in $B$ with unique upper cover $b^\uparrow \in B$. Write factor tables $\psi_c \propto e^{-H_c}$ on $B$. For $x \in E_b$ and any $\mathbf{k} \in E_{b^\uparrow}$ with $\mathbf{k}|_b = x$,*
$$\psi'_{b^\uparrow}(\mathbf{k}) \propto \psi_{b^\uparrow}(\mathbf{k}) \cdot \frac{\displaystyle\sum_{z \in E_a:\, z_b = x} \psi_a(z)}{\psi_b(x)}$$

*and $\psi'_c = \psi_c$ for all $c \neq b^\uparrow$.*

**Lemma D.4** (Factor version of S–P Eq. 4.30: colinear, survivor not linear). *Let $a \in \mathcal{A}$ be colinear and maximal, $B = \mathcal{A} \setminus \{a\}$, $b = a^\downarrow \in B$ not* linear *in $B$, and let $c_B(b)$ be the Möbius coefficient of $b$ in the poset $B$. Then*

$$\psi'_b(x) \propto \big(\psi_b(x)\big)^{1 - \frac{1}{c_B(b)}} \left( \sum_{z \in E_a:\, z_b = x} \psi_a(z) \right)^{\frac{1}{c_B(b)}} \qquad (x \in E_b),$$

*and $\psi'_c = \psi_c$ for all $c \neq b$.*

*Both lemmas.* Start from the $H$-space statements in S–P and exponentiate. For Eq. 4.29: $\widetilde{H}_{b^\uparrow} = H_{b^\uparrow} - \widehat{H}_b \circ \pi_{b^\uparrow b}$ with $\widehat{H}_b(x) = \ln \sum_{z:\, z_b = x} e^{-H_a(z) + H_b(x)}$ gives

$$\psi'_{b^\uparrow}(\mathbf{k}) \propto e^{-\widetilde{H}_{b^\uparrow}(\mathbf{k})} \propto e^{-H_{b^\uparrow}(\mathbf{k})}\, e^{\widehat{H}_b(x)} = \psi_{b^\uparrow}(\mathbf{k}) \frac{\sum_{z:\, z_b = x} e^{-H_a(z)}}{e^{-H_b(x)}} = \psi_{b^\uparrow}(\mathbf{k}) \frac{\sum_{z:\, z_b = x} \psi_a(z)}{\psi_b(x)}.$$

For Eq. 4.30: $\widetilde{H}_b = H_b - \frac{1}{c_B(b)} \ln \sum_{z:\, z_b = x} e^{-H_a(z) + H_b(x)}$ yields

$$\psi_b'(x) \;\propto\; e^{-\widetilde{H}_b(x)} = \psi_b(x) \left( \sum_{z:\, z_b = x} e^{-H_a(z) + H_b(x)} \right)^{1/c_B(b)} = \psi_b(x) \left( \frac{1}{\psi_b(x)} \sum_{z:\, z_b = x} \psi_a(z) \right)^{1/c_B(b)},$$

we can verify this is correct by taking logs. $\qquad\square$

*Proof of Prop 6.2.* We verify the two functor axioms: preservation of identities and preservation of composition.

**(Identities).** If $r = \mathrm{id}_{(A,H)}$, then no poset point is retracted and the Hamiltonians $H_c$ (hence the tables $\psi_c$) are unchanged. The compilation on objects produces the same CRN $(X, R, \kappa)$, and on morphisms we have

$$\varphi_0 = \mathrm{id}_X, \quad \varphi_1 = \mathrm{id}_R, \quad \psi = \mathrm{id}_\kappa,$$

so $\mathcal{N}(\mathrm{id}_{(A,H)}) = \mathrm{id}_{\mathcal{N}(A,H)}$.

**(Composition).** Let

$$r_1 : (A, H) \to (B, G), \quad r_2 : (B, G) \to (C, K)$$

be two morphisms in **FGraphRet**. Their composite $r_2 \circ r_1$ is again a finite composite of linear/colinear retractions. On factor tables, the SP–B update rules are defined by local formulas in $\psi$-space; applying $r_1$ and then $r_2$ gives the same final tables $\psi_c''$ as applying the composite $r_2 \circ r_1$ once.

On CRNs we have:

- $\varphi_0^{r_2 \circ r_1}$ deletes exactly the union of bundles deleted by $r_1$ and $r_2$; this is the same as $\varphi_0^{r_2} \circ \varphi_0^{r_1}$ bundlewise.

- Likewise, $\varphi_1^{r_2 \circ r_1} = \varphi_1^{r_2} \circ \varphi_1^{r_1}$ because a reaction is present after the composite if and only if all its bundles survive both stages.

- For the rate part, each surviving edge $j \to n$ carries a scalar $c_{j \to n}$ that we keep fixed along the entire chain. If $\psi_j \mapsto \psi_j' \mapsto \psi_j''$ are the factor tables under $r_1$ and $r_2$, then

$$\lambda_{j \to n} = c_{j \to n} \psi_j \xmapsto{\psi^{r_1}} c_{j \to n} \psi_j' \xmapsto{\psi^{r_2}} c_{j \to n} \psi_j'',$$

  which is exactly the update prescribed by $\psi^{r_2 \circ r_1}$. Hence $\psi^{r_2 \circ r_1} = \psi^{r_2} \circ \psi^{r_1}$.

Therefore

$$\mathcal{N}(r_2 \circ r_1) = (\varphi_0^{r_2 \circ r_1}, \varphi_1^{r_2 \circ r_1}, \psi^{r_2 \circ r_1}) = (\varphi_0^{r_2}, \varphi_1^{r_2}, \psi^{r_2}) \circ (\varphi_0^{r_1}, \varphi_1^{r_1}, \psi^{r_1}) = \mathcal{N}(r_2) \circ \mathcal{N}(r_1).$$

**(Closure under (W1)–(W6)).** For any factor graph $(A, H)$, the construction of $\mathcal{N}(A, H)$ forces the bundle decomposition and zero$\leftrightarrow$coordinate structure, so (W1)–(W3) always hold.

We verify (W4)–(W6) directly:

- **(W4):** By construction, sum-bundle production reactions have the form

$$S_0^{j \to n} + \sum_{n' \neq n} P_{k_{n'}^j}^{n' \to j} \to S_k^{j \to n} + \sum_{n' \neq n} P_{k_{n'}^j}^{n' \to j},$$

  with rate $\psi_j(\mathbf{k}^j)$. The catalytic neighbors of $S^{j \to n}$ are the product bundles $\{P^{n' \to j}\}_{n' \in \mathrm{ne}(j) \setminus n}$, and each monomial uses exactly one coordinate from each such bundle. This is precisely the assignment-form expansion required by (W4).

- **(W5):** Product bundles have production reactions of the form

$$P_0^{n\to j} + \sum_{j'\neq j} S_k^{j'\to n} \to P_k^{n\to j} + \sum_{j'\neq j} S_k^{j'\to n}.$$

  For each $k$, there is exactly one assignment (all catalysts contribute state $k$), and the bijections $\theta_{P,B}$ are simply the identity on the state set $E_n$. This satisfies (W5).

- **(W6):** Sum bundles for the same factor $j$ form a factor class $J_j$. For each variable $n \in \mathrm{ne}(j)$, the sum bundle $S^{j\to n}$ has $\mathrm{Cat}(S^{j\to n}) = \{P^{n'\to j} : n' \neq n\} = V_{J_j} \setminus \{P^{n\to j}\}$, establishing the missing-one-neighbor pattern. The coefficient arrays are $C_{S^{j\to n}}(k;a) = \psi_j(x_n = k, x_{n'} = a(n'))$, which is the clamped-table property with $\kappa_{j\to n} = 1$.

SP–B retractions only change the tables $\psi_J$ within a factor class and do not permute or merge catalytic bundles. Thus (W4)–(W6) continue to hold after any sequence of retractions, and the target CRN of $\mathcal{N}(r)$ still lies in the subcategory of CRNs satisfying (W1)–(W6).

Altogether, $\mathcal{N}$ preserves identities and composition, and by Lemma 6.1 its image is contained in the CRNs satisfying (W1)–(W6) and (R1), as claimed. $\qquad\square$

*Proof of Theorem 6.3.* By Lemma 6.1, both $\Gamma = \mathcal{N}(A, H)$ and $\Gamma' = \mathcal{N}(A', H')$ satisfy (W1)–(W6) and (R1). By Theorem 5.4, their positive steady states correspond to BP fixed points on the respective factor graphs.

This is exactly the definition of $\mathcal{N}(r)$ given above. The key points are:

- $\varphi_0$ deletes bundles corresponding to removed edges/variables;

- $\varphi_1$ deletes reactions involving those bundles;

- $\psi$ updates production rates according to the SP–B table update, while leaving recycling rates unchanged.

The main theorem of Sergeant–Perthuis–Boitel (Sergeant-Perthuis & Boitel, 2025) shows that SP–B retractions $r : (A, H) \to (A', H')$ map BP fixed points on $(A, H)$ to BP fixed points on $(A', H')$ by restricting to the surviving sites and updating factor tables. We verify that this correspondence is reflected at the CRN level.

Let $\pi$ be a BP fixed point on $(A, H)$. By Theorem 5.4, there exists a positive steady state $x^*$ of $\Gamma = \mathcal{N}(A, H)$ such that the bundle-normalized concentrations

$$\frac{[B_k]^*}{\sum_{k'}[B_{k'}]^*}$$

equal the corresponding message components of $\pi$.

Let $\pi'$ be the restriction of $\pi$ to $(A', H')$ as defined by the SP–B retraction. The Sergeant–Perthuis–Boitel theorem guarantees that $\pi'$ is a BP fixed point on $(A', H')$.

Now consider the CRN morphism $\mathcal{N}(r) = (\varphi_0, \varphi_1, \psi)$. For surviving bundles $B$, the morphism:

- Preserves the bundle structure (same zero and coordinate species);

- Preserves recycling rates ($\rho_B$ unchanged);

- Updates production rates according to $\psi_j \mapsto \psi'_j$.

The steady-state equations for surviving bundles in $\Gamma'$ are:

$$\frac{\rho_B}{[B_0]'}[B_k]' = \kappa_{j\to n} \sum_a \psi'_j(\mathbf{k}^j) \prod_{Q'\in V_J\setminus\{Q\}} [Q'_{a(Q')}]'.$$

Since the factor-table update $\psi_j \mapsto \psi'_j$ is exactly what appears in the SP–B prescription, and since $\pi'$ satisfies the BP equations with tables $\psi'_j$, there exists a positive steady state $x'^*$ of $\Gamma'$ whose bundle-normalized concentrations equal the message components of $\pi'$.

The relationship between $x^*$ and $x'^*$ is that their restrictions to surviving bundles are related by the concentration rescaling induced by $\mathcal{N}(r)$. Specifically, if $B$ is a surviving bundle, then

$$[B_k]'^* = s_B \cdot [B_k]^*$$

for some positive scalar $s_B$ that depends on the normalization conventions but preserves the bundle-normalized ratios.

This completes the proof that $\mathcal{N}(r)$ transports BP fixed points from $\Gamma$ to $\Gamma'$. $\qquad\square$

## E. Examples

These examples illustrate the two main concrete mechanisms behind the paper. Example E.1 shows that the steady-state equations of the compiled CRN reproduce the BP fixed-point equations on a small mixed-cardinality chain. Example E.2 then shows how an SP–B retraction acts on the same instance and how the induced CRN morphism removes the corresponding bundles while preserving the BP semantics on the surviving variables.

*Example* E.1 (Chain graph with mixed cardinalities). Consider variables $\{1, 2, 3\}$ with state spaces $E_1 = \{1, 2\}$, $E_2 = \{1, 2, 3\}$, $E_3 = \{1, 2\}$, and factors $a, b$ with $\mathrm{ne}(a) = \{1, 2\}$, $\mathrm{ne}(b) = \{2, 3\}$. The factor tables are

$$\psi_a : E_1 \times E_2 \to (0, \infty), \qquad \psi_b : E_2 \times E_3 \to (0, \infty),$$

so $\psi_a$ is $2 \times 3$ and $\psi_b$ is $3 \times 2$.

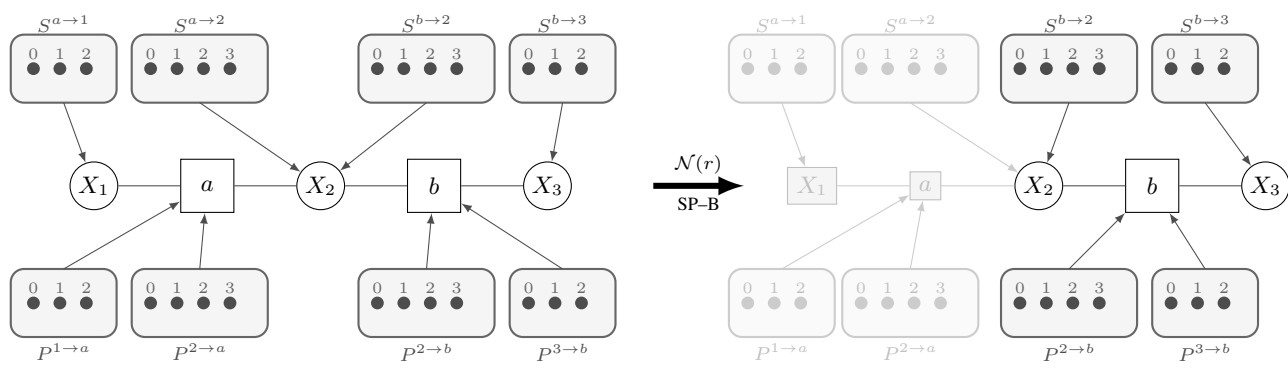

**(a) Full CRN $\Gamma$: 28 species**    **(b) Reduced CRN $\Gamma'$: 14 species**

*Figure 6.* Bundle structure of the compiled CRN $\Gamma = \mathcal{N}(\mathcal{F})$ (left) and the reduced CRN $\Gamma' = \mathcal{N}(\mathcal{F}')$ (right) for the mixed-cardinality chain of Example E.1 and its reduction in Example E.2. Each directed edge of the factor graph produces one sum bundle (above) and one product bundle (below); dots represent individual chemical species (zero species indexed 0, coordinate species indexed $1, \ldots, |E|$). The SP–B retraction $r$ removes variable $X_1$ and factor $a$, deleting the four faded bundles and their associated reactions. The surviving bundles carry the same BP beliefs on $X_2$ and $X_3$ as the full network, with the influence of $X_1$ absorbed into the updated rate parameters of $S^{b\to2}$ and $S^{b\to3}$.

**Bundles.** Each directed edge of the factor graph produces one bundle. The four sum bundles and four product bundles are:

| Bundle | Type | $|E_B|$ | Target/Source | Species |
|--------|------|---------|---------------|---------|
| $S^{a\to1}$ | sum | 2 | Tgt $= P^{1\to a}$ | $S_0^{a\to1}$, $S_1^{a\to1}$, $S_2^{a\to1}$ |
| $S^{a\to2}$ | sum | 3 | Tgt $= P^{2\to a}$ | $S_0^{a\to2}$, $S_1^{a\to2}$, $S_2^{a\to2}$, $S_3^{a\to2}$ |
| $S^{b\to2}$ | sum | 3 | Tgt $= P^{2\to b}$ | $S_0^{b\to2}$, $S_1^{b\to2}$, $S_2^{b\to2}$, $S_3^{b\to2}$ |
| $S^{b\to3}$ | sum | 2 | Tgt $= P^{3\to b}$ | $S_0^{b\to3}$, $S_1^{b\to3}$, $S_2^{b\to3}$ |
| $P^{1\to a}$ | product | 2 | var $1 \to$ fac $a$ | $P_0^{1\to a}$, $P_1^{1\to a}$, $P_2^{1\to a}$ |
| $P^{2\to a}$ | product | 3 | var $2 \to$ fac $a$ | $P_0^{2\to a}$, $P_1^{2\to a}$, $P_2^{2\to a}$, $P_3^{2\to a}$ |
| $P^{2\to b}$ | product | 3 | var $2 \to$ fac $b$ | $P_0^{2\to b}$, $P_1^{2\to b}$, $P_2^{2\to b}$, $P_3^{2\to b}$ |
| $P^{3\to b}$ | product | 2 | var $3 \to$ fac $b$ | $P_0^{3\to b}$, $P_1^{3\to b}$, $P_2^{3\to b}$ |

Here subscript 0 is the zero (unassigned) species $X_{B,0}$, and subscripts $1, \ldots, |E_B|$ are the coordinate species. Note that bundles associated with variable 2 have $|E_B| = 3$ coordinate species, while those for variables 1 or 3 have $|E_B| = 2$.

**Catalytic neighbor sets.** For each sum bundle $B$, $\mathrm{Cat}(B)$ consists of the product bundles in the same factor class minus the target (the missing-one-neighbor pattern of (W6.1)):

$$\mathrm{Cat}(S^{a\to1}) = V_{J_a} \setminus \{P^{1\to a}\} = \{P^{2\to a}\},$$
$$\mathrm{Cat}(S^{a\to2}) = V_{J_a} \setminus \{P^{2\to a}\} = \{P^{1\to a}\},$$
$$\mathrm{Cat}(S^{b\to2}) = V_{J_b} \setminus \{P^{2\to b}\} = \{P^{3\to b}\},$$
$$\mathrm{Cat}(S^{b\to3}) = V_{J_b} \setminus \{P^{3\to b}\} = \{P^{2\to b}\}.$$

For each product bundle $P$, $\mathrm{Cat}(P)$ consists of the sum bundles targeting the same variable from *other* factors:

$$\mathrm{Cat}(P^{1\to a}) = \varnothing \qquad \text{(variable 1 touches only factor } a\text{)},$$
$$\mathrm{Cat}(P^{2\to a}) = \{S^{b\to2}\} \qquad \text{(sum bundle from factor } b \text{ targeting var 2)},$$
$$\mathrm{Cat}(P^{2\to b}) = \{S^{a\to2}\} \qquad \text{(sum bundle from factor } a \text{ targeting var 2)},$$
$$\mathrm{Cat}(P^{3\to b}) = \varnothing \qquad \text{(variable 3 touches only factor } b\text{)}.$$

**Factor classes and variable classes.**

- $J_a = \{S^{a\to 1}, S^{a\to 2}\}$ with $V_{J_a} = \{P^{1\to a}, P^{2\to a}\}$; factor table $\psi_a : E_1 \times E_2 \to (0, \infty)$, a $2 \times 3$ matrix.

- $J_b = \{S^{b\to 2}, S^{b\to 3}\}$ with $V_{J_b} = \{P^{2\to b}, P^{3\to b}\}$; factor table $\psi_b : E_2 \times E_3 \to (0, \infty)$, a $3 \times 2$ matrix.

The cross-link relation $\sim$ on product bundles (Definition 5.1) gives: $P^{2\to a} \sim P^{2\to b}$ via $S^{b\to 2}$ (target $P^{2\to b}$, catalyzes $P^{2\to a}$) and symmetrically via $S^{a\to 2}$. No other pairs are cross-linked. The variable classes are therefore

$$\{P^{1\to a}\}, \quad \{P^{2\to a}, P^{2\to b}\}, \quad \{P^{3\to b}\},$$

with cardinalities $2, 3, 2$, matching variables $1, 2, 3$.

**Napp compilation $\mathcal{N}(\mathcal{F})$: species.** Each bundle $B$ with $|E_B|$ coordinate species contributes $|E_B| + 1$ species (one zero species $+ |E_B|$ coordinate species). The total species count is

$$\underbrace{(2{+}1)}_{S^{a\to 1}} + \underbrace{(3{+}1)}_{S^{a\to 2}} + \underbrace{(3{+}1)}_{S^{b\to 2}} + \underbrace{(2{+}1)}_{S^{b\to 3}} + \underbrace{(2{+}1)}_{P^{1\to a}} + \underbrace{(3{+}1)}_{P^{2\to a}} + \underbrace{(3{+}1)}_{P^{2\to b}} + \underbrace{(2{+}1)}_{P^{3\to b}} = \textbf{28 species}.$$

**Recycling reactions.** For each bundle $B$ and each coordinate species $B_k$, $k \geq 1$ (Definition D.1, step 2):

$$B_k \xrightarrow{\kappa_r} B_0.$$

The number of recycling reactions per bundle equals $|E_B|$, giving $2 + 3 + 3 + 2 + 2 + 3 + 3 + 2 = \textbf{20}$ recycling reactions. We list them explicitly:

*Sum-bundle recycling (10 reactions):*

$$S_1^{a\to 1} \xrightarrow{\kappa_r} S_0^{a\to 1}, \qquad S_2^{a\to 1} \xrightarrow{\kappa_r} S_0^{a\to 1},$$
$$S_1^{a\to 2} \xrightarrow{\kappa_r} S_0^{a\to 2}, \qquad S_2^{a\to 2} \xrightarrow{\kappa_r} S_0^{a\to 2}, \quad S_3^{a\to 2} \xrightarrow{\kappa_r} S_0^{a\to 2},$$
$$S_1^{b\to 2} \xrightarrow{\kappa_r} S_0^{b\to 2}, \qquad S_2^{b\to 2} \xrightarrow{\kappa_r} S_0^{b\to 2}, \quad S_3^{b\to 2} \xrightarrow{\kappa_r} S_0^{b\to 2},$$
$$S_1^{b\to 3} \xrightarrow{\kappa_r} S_0^{b\to 3}, \qquad S_2^{b\to 3} \xrightarrow{\kappa_r} S_0^{b\to 3}.$$

*Product-bundle recycling (10 reactions):*

$$P_1^{1\to a} \xrightarrow{\kappa_r} P_0^{1\to a}, \qquad P_2^{1\to a} \xrightarrow{\kappa_r} P_0^{1\to a},$$
$$P_1^{2\to a} \xrightarrow{\kappa_r} P_0^{2\to a}, \qquad P_2^{2\to a} \xrightarrow{\kappa_r} P_0^{2\to a}, \quad P_3^{2\to a} \xrightarrow{\kappa_r} P_0^{2\to a},$$
$$P_1^{2\to b} \xrightarrow{\kappa_r} P_0^{2\to b}, \qquad P_2^{2\to b} \xrightarrow{\kappa_r} P_0^{2\to b}, \quad P_3^{2\to b} \xrightarrow{\kappa_r} P_0^{2\to b},$$
$$P_1^{3\to b} \xrightarrow{\kappa_r} P_0^{3\to b}, \qquad P_2^{3\to b} \xrightarrow{\kappa_r} P_0^{3\to b}.$$

**Sum-bundle production reactions.** For each directed edge $j \to n$ and each joint assignment $k^j \in \prod_{n' \in \mathrm{ne}(j)} E_{n'}$ with $(k^j)_n = k$ (Definition D.1, step 3):

$$S_0^{j\to n} + \sum_{n' \in \mathrm{ne}(j)\setminus n} P_{(k^j)_{n'}}^{n'\to j} \xrightarrow{\psi_j(k^j)} S_k^{j\to n} + \sum_{n' \in \mathrm{ne}(j)\setminus n} P_{(k^j)_{n'}}^{n'\to j}.$$

**Sum bundle $S^{a\to 1}$** (target: variable 1; catalyst: $P^{2\to a}$). The assignment is $k^a = (k_1, k_2) \in E_1 \times E_2$ with output state $k = k_1$. There are $|E_1| \times |E_2| = 2 \times 3 = 6$ reactions, one per entry of $\psi_a$:

| $(k_1, k_2)$ | Reaction | Rate |
|---|---|---|
| $(1,1)$ | $S_0^{a \to 1} + P_1^{2 \to a} \to S_1^{a \to 1} + P_1^{2 \to a}$ | $\psi_a(1,1)$ |
| $(1,2)$ | $S_0^{a \to 1} + P_2^{2 \to a} \to S_1^{a \to 1} + P_2^{2 \to a}$ | $\psi_a(1,2)$ |
| $(1,3)$ | $S_0^{a \to 1} + P_3^{2 \to a} \to S_1^{a \to 1} + P_3^{2 \to a}$ | $\psi_a(1,3)$ |
| $(2,1)$ | $S_0^{a \to 1} + P_1^{2 \to a} \to S_2^{a \to 1} + P_1^{2 \to a}$ | $\psi_a(2,1)$ |
| $(2,2)$ | $S_0^{a \to 1} + P_2^{2 \to a} \to S_2^{a \to 1} + P_2^{2 \to a}$ | $\psi_a(2,2)$ |
| $(2,3)$ | $S_0^{a \to 1} + P_3^{2 \to a} \to S_2^{a \to 1} + P_3^{2 \to a}$ | $\psi_a(2,3)$ |

Observe that $P^{2 \to a}$ species appear on both sides (zero net stoichiometry)—they are catalysts. The net stoichiometry of each reaction is $S_0^{a \to 1} \to S_k^{a \to 1}$, supported entirely in the bundle $S^{a \to 1}$, verifying (W1) and (W3).

**Sum bundle** $S^{a \to 2}$ (target: variable 2; catalyst: $P^{1 \to a}$). Assignment $k^a = (k_1, k_2)$ with output $k = k_2$. There are $2 \times 3 = 6$ reactions:

| $(k_1, k_2)$ | Reaction | Rate |
|---|---|---|
| $(1,1)$ | $S_0^{a \to 2} + P_1^{1 \to a} \to S_1^{a \to 2} + P_1^{1 \to a}$ | $\psi_a(1,1)$ |
| $(2,1)$ | $S_0^{a \to 2} + P_2^{1 \to a} \to S_1^{a \to 2} + P_2^{1 \to a}$ | $\psi_a(2,1)$ |
| $(1,2)$ | $S_0^{a \to 2} + P_1^{1 \to a} \to S_2^{a \to 2} + P_1^{1 \to a}$ | $\psi_a(1,2)$ |
| $(2,2)$ | $S_0^{a \to 2} + P_2^{1 \to a} \to S_2^{a \to 2} + P_2^{1 \to a}$ | $\psi_a(2,2)$ |
| $(1,3)$ | $S_0^{a \to 2} + P_1^{1 \to a} \to S_3^{a \to 2} + P_1^{1 \to a}$ | $\psi_a(1,3)$ |
| $(2,3)$ | $S_0^{a \to 2} + P_2^{1 \to a} \to S_3^{a \to 2} + P_2^{1 \to a}$ | $\psi_a(2,3)$ |

Here the catalyst $P^{1 \to a}$ has only $|E_1| = 2$ coordinate species, while the target bundle $S^{a \to 2}$ has $|E_{S^{a \to 2}}| = 3$ coordinate species. This is where mixed cardinalities become visible in the reaction stoichiometry: the number of reactions producing a given coordinate $S_k^{a \to 2}$ equals $|E_1| = 2$ (one per state of the catalyst), whereas each reaction producing $S_k^{a \to 1}$ (above) had $|E_2| = 3$ such variants.

**Sum bundle** $S^{b \to 2}$ (target: variable 2; catalyst: $P^{3 \to b}$). Assignment $k^b = (k_2, k_3)$ with output $k = k_2$. There are $3 \times 2 = 6$ reactions:

| $(k_2, k_3)$ | Reaction | Rate |
|---|---|---|
| $(1,1)$ | $S_0^{b \to 2} + P_1^{3 \to b} \to S_1^{b \to 2} + P_1^{3 \to b}$ | $\psi_b(1,1)$ |
| $(1,2)$ | $S_0^{b \to 2} + P_2^{3 \to b} \to S_1^{b \to 2} + P_2^{3 \to b}$ | $\psi_b(1,2)$ |
| $(2,1)$ | $S_0^{b \to 2} + P_1^{3 \to b} \to S_2^{b \to 2} + P_1^{3 \to b}$ | $\psi_b(2,1)$ |
| $(2,2)$ | $S_0^{b \to 2} + P_2^{3 \to b} \to S_2^{b \to 2} + P_2^{3 \to b}$ | $\psi_b(2,2)$ |
| $(3,1)$ | $S_0^{b \to 2} + P_1^{3 \to b} \to S_3^{b \to 2} + P_1^{3 \to b}$ | $\psi_b(3,1)$ |
| $(3,2)$ | $S_0^{b \to 2} + P_2^{3 \to b} \to S_3^{b \to 2} + P_2^{3 \to b}$ | $\psi_b(3,2)$ |

**Sum bundle** $S^{b \to 3}$ (target: variable 3; catalyst: $P^{2 \to b}$). Assignment $k^b = (k_2, k_3)$ with output $k = k_3$. There are $3 \times 2 = 6$ reactions:

| $(k_2, k_3)$ | Reaction | Rate |
|---|---|---|
| $(1, 1)$ | $S_0^{b \to 3} + P_1^{2 \to b} \to S_1^{b \to 3} + P_1^{2 \to b}$ | $\psi_b(1, 1)$ |
| $(2, 1)$ | $S_0^{b \to 3} + P_2^{2 \to b} \to S_1^{b \to 3} + P_2^{2 \to b}$ | $\psi_b(2, 1)$ |
| $(3, 1)$ | $S_0^{b \to 3} + P_3^{2 \to b} \to S_1^{b \to 3} + P_3^{2 \to b}$ | $\psi_b(3, 1)$ |
| $(1, 2)$ | $S_0^{b \to 3} + P_1^{2 \to b} \to S_2^{b \to 3} + P_1^{2 \to b}$ | $\psi_b(1, 2)$ |
| $(2, 2)$ | $S_0^{b \to 3} + P_2^{2 \to b} \to S_2^{b \to 3} + P_2^{2 \to b}$ | $\psi_b(2, 2)$ |
| $(3, 2)$ | $S_0^{b \to 3} + P_3^{2 \to b} \to S_2^{b \to 3} + P_3^{2 \to b}$ | $\psi_b(3, 2)$ |

**Sum-bundle production count:** $6 + 6 + 6 + 6 = \mathbf{24}$ reactions.

**Product-bundle production reactions.** For each variable $n$, each factor $j \in \mathrm{ne}(n)$, and each state $k \in E_n$ (Definition D.1, step 4):

$$P_0^{n \to j} + \sum_{j' \in \mathrm{ne}(n) \setminus j} S_k^{j' \to n} \xrightarrow{\kappa_{\mathrm{prod}}} P_k^{n \to j} + \sum_{j' \in \mathrm{ne}(n) \setminus j} S_k^{j' \to n}.$$

When $|\mathrm{ne}(n) \setminus j| = 0$ (i.e. the variable touches only one factor), there are no catalysts and the reaction is simply $P_0^{n \to j} \xrightarrow{\kappa_{\mathrm{prod}}} P_k^{n \to j}$.

**Product bundle $P^{1 \to a}$** ($\mathrm{ne}(1) = \{a\}$, so $\mathrm{ne}(1) \setminus a = \varnothing$; no catalysts). Two reactions ($|E_1| = 2$):

$$P_0^{1 \to a} \xrightarrow{\kappa_{\mathrm{prod}}} P_1^{1 \to a}, \qquad P_0^{1 \to a} \xrightarrow{\kappa_{\mathrm{prod}}} P_2^{1 \to a}.$$

**Product bundle $P^{2 \to a}$** ($\mathrm{ne}(2) = \{a, b\}$, so $\mathrm{ne}(2) \setminus a = \{b\}$; catalyst: $S^{b \to 2}$). Three reactions ($|E_2| = 3$):

$$P_0^{2 \to a} + S_1^{b \to 2} \xrightarrow{\kappa_{\mathrm{prod}}} P_1^{2 \to a} + S_1^{b \to 2},$$
$$P_0^{2 \to a} + S_2^{b \to 2} \xrightarrow{\kappa_{\mathrm{prod}}} P_2^{2 \to a} + S_2^{b \to 2},$$
$$P_0^{2 \to a} + S_3^{b \to 2} \xrightarrow{\kappa_{\mathrm{prod}}} P_3^{2 \to a} + S_3^{b \to 2}.$$

Here the bijection $\theta_{P^{2 \to a}, S^{b \to 2}}$ from (W5) is the identity on $E_2 = \{1, 2, 3\}$: coordinate $k$ of the catalyst $S^{b \to 2}$ produces coordinate $k$ of $P^{2 \to a}$. Both bundles have cardinality 3 because both are associated with variable 2.

**Product bundle $P^{2 \to b}$** ($\mathrm{ne}(2) \setminus b = \{a\}$; catalyst: $S^{a \to 2}$). Three reactions:

$$P_0^{2 \to b} + S_1^{a \to 2} \xrightarrow{\kappa_{\mathrm{prod}}} P_1^{2 \to b} + S_1^{a \to 2},$$
$$P_0^{2 \to b} + S_2^{a \to 2} \xrightarrow{\kappa_{\mathrm{prod}}} P_2^{2 \to b} + S_2^{a \to 2},$$
$$P_0^{2 \to b} + S_3^{a \to 2} \xrightarrow{\kappa_{\mathrm{prod}}} P_3^{2 \to b} + S_3^{a \to 2}.$$

**Product bundle $P^{3 \to b}$** ($\mathrm{ne}(3) = \{b\}$, so no catalysts). Two reactions:

$$P_0^{3 \to b} \xrightarrow{\kappa_{\mathrm{prod}}} P_1^{3 \to b}, \qquad P_0^{3 \to b} \xrightarrow{\kappa_{\mathrm{prod}}} P_2^{3 \to b}.$$

**Product-bundle production count:** $2 + 3 + 3 + 2 = \mathbf{10}$ reactions.

**CRN summary.**

| | |
|---|---|
| Species: | 28 |
| Recycling reactions: | 20 |
| Sum-bundle production reactions: | 24 |
| Product-bundle production reactions: | 10 |
| Total reactions: | **54** |

**Verification of the recognition conditions.** We verify that the compiled CRN $\mathcal{N}(\mathcal{F})$ satisfies (W1)–(W6) and (R1).

(W1) Every reaction has net stoichiometry supported in a single bundle. Recycling reactions change only $B_k \to B_0$ within their bundle. Sum- and product-production reactions convert $B_0 \to B_k$ within a single bundle; catalytic species appear on both sides with zero net change.

(W2) Each bundle has a distinguished zero species ($S_0^{a \to 1}$, $P_0^{1 \to a}$, etc.).

(W3) All internal net stoichiometries are of the form $-X_{B,0} + X_{B,k}$ (production) or $X_{B,0} - X_{B,k}$ (recycling).

(W4) Consider $S^{a \to 1}$ with $\mathrm{Cat}(S^{a \to 1}) = \{P^{2 \to a}\}$. Its positive production is

$$\frac{d}{dt}[S_k^{a \to 1}]^+ = [S_0^{a \to 1}] \sum_{k_2 \in E_2} \psi_a(k, k_2)\,[P_{k_2}^{2 \to a}],$$

which is multilinear in the catalytic coordinates, with assignment table $C_{S^{a \to 1}}(k;\, k_2) = \psi_a(k, k_2)$, confirming the separability requirement. The other sum bundles are analogous.

(W5) Consider $P^{2 \to a}$ with $\mathrm{Cat}(P^{2 \to a}) = \{S^{b \to 2}\}$. For each $k \in E_2$, the unique supporting assignment is $a^{(k)}(S^{b \to 2}) = k$ (the identity bijection), and the rate is $\kappa_{\mathrm{prod}}$ independent of $k$. This is the diagonal/bijection structure. The endpoint bundles $P^{1 \to a}$ and $P^{3 \to b}$ have $\mathrm{Cat} = \varnothing$, so (W5) holds vacuously.

(W6) The partition of sum bundles into factor classes is $J_a = \{S^{a \to 1}, S^{a \to 2}\}$, $J_b = \{S^{b \to 2}, S^{b \to 3}\}$. For $J_a$: $|J_a| = |V_{J_a}| = 2$; $\mathrm{Cat}(S^{a \to 1}) = V_{J_a} \setminus \{P^{1 \to a}\} = \{P^{2 \to a}\}$ (missing-one-neighbor). The state-space identification $\iota_{J_a \to P^{1 \to a}} : E_1 \to E_{S^{a \to 1}}$ is the identity, and $\iota_{J_a \to P^{2 \to a}} : E_2 \to E_{S^{a \to 2}}$ is the identity, confirming (W6.2) with $|E_1| = 2 \neq 3 = |E_2|$—the mixed-cardinality case. The clamped-table property (W6.3) holds: $C_{S^{a \to 1}}(k;\, k_2) = \psi_a(k, k_2)$ as computed above.

(R1) All recycling reactions use the global rate $\kappa_r$, and all product-production reactions use the global rate $\kappa_{\mathrm{prod}}$, both independent of the state $k$.

**Steady-state equations (Theorem 5.4).** At a positive steady state of $\mathcal{N}(\mathcal{F})$, the sum-bundle concentrations satisfy

$$\frac{\kappa_r}{[S_0^{a \to 1}]}\,[S_k^{a \to 1}] = \sum_{k_2=1}^{3} \psi_a(k, k_2)\,[P_{k_2}^{2 \to a}], \qquad\qquad k \in \{1, 2\}, \tag{7}$$

$$\frac{\kappa_r}{[S_0^{a \to 2}]}\,[S_k^{a \to 2}] = \sum_{k_1=1}^{2} \psi_a(k_1, k)\,[P_{k_1}^{1 \to a}], \qquad\qquad k \in \{1, 2, 3\}, \tag{8}$$

$$\frac{\kappa_r}{[S_0^{b \to 2}]}\,[S_k^{b \to 2}] = \sum_{k_3=1}^{2} \psi_b(k, k_3)\,[P_{k_3}^{3 \to b}], \qquad\qquad k \in \{1, 2, 3\}, \tag{9}$$

$$\frac{\kappa_r}{[S_0^{b \to 3}]}\,[S_k^{b \to 3}] = \sum_{k_2=1}^{3} \psi_b(k_2, k)\,[P_{k_2}^{2 \to b}], \qquad\qquad k \in \{1, 2\}, \tag{10}$$

and the product-bundle concentrations satisfy

$$\frac{\kappa_r}{\kappa_{\mathrm{prod}}\,[P_0^{1 \to a}]}\,[P_k^{1 \to a}] = 1, \qquad\qquad k \in \{1, 2\}, \tag{11}$$

$$\frac{\kappa_r}{\kappa_{\mathrm{prod}}\,[P_0^{2 \to a}]}\,[P_k^{2 \to a}] = [S_k^{b \to 2}], \qquad\qquad k \in \{1, 2, 3\}, \tag{12}$$

$$\frac{\kappa_r}{\kappa_{\mathrm{prod}}\,[P_0^{2 \to b}]}\,[P_k^{2 \to b}] = [S_k^{a \to 2}], \qquad\qquad k \in \{1, 2, 3\}, \tag{13}$$

$$\frac{\kappa_r}{\kappa_{\mathrm{prod}}\,[P_0^{3 \to b}]}\,[P_k^{3 \to b}] = 1, \qquad\qquad k \in \{1, 2\}. \tag{14}$$

Equations (11) and (14) say that $P^{1\to a}$ and $P^{3\to b}$ are uniform at steady state (no incoming sum messages, since variables 1 and 3 each touch only one factor). Equations (12) and (13) express the product-message update for variable 2: the message from variable 2 to factor $a$ is proportional to the sum message arriving from factor $b$, and vice versa, matching the BP product-message equation (9) with $\mathrm{ne}(2) \setminus a = \{b\}$.

Together, equations (7)–(14) are exactly the sum-product BP fixed-point equations on the chain $1$–$a$–$2$–$b$–$3$ with mixed cardinalities $(|E_1|, |E_2|, |E_3|) = (2, 3, 2)$, confirming Theorem 5.4 on this instance.

**Remark (effect of mixed cardinalities on CRN size).** In the uniform-cardinality case $|E_i| = K$ for all $i$, a chain on $n$ variables produces $4(n-1)$ bundles each of size $K+1$, giving $4(n-1)(K+1)$ species and $4(n-1)K + (n-1)K^2 + (n-1)K^2$ reactions (recycling + sum-production + product-production). With mixed cardinalities, these counts depend on the per-variable $|E_i|$ and the per-factor product $\prod_{i\in\mathrm{ne}(j)} |E_i|$. For our instance, the asymmetry $|E_2| = 3$ versus $|E_1| = |E_3| = 2$ is visible in the reaction tables above: factor $a$ generates $|E_1| \cdot |E_2| = 6$ sum-production reactions per sum bundle, while a hypothetical uniform binary chain would generate $2 \cdot 2 = 4$.

The example below shows that the reduction deletes entire message bundles associated with retractable tendrils while leaving the core computation unchanged. On this small instance, the reduced CRN therefore has fewer species and reactions but still computes the same BP information on the surviving variables.

*Example* E.2 (Reduction of the mixed-cardinality chain). We continue Example E.1 and apply SP–B retractions to the chain $1$–$a$–$2$–$b$–$3$, tracking the induced CRN morphism $\mathcal{N}(r)$ at each step.

**Poset structure.** The associated poset $A$ has five elements $\{1, 2, 3, a, b\}$ with cover relations $1 < a$, $2 < a$, $2 < b$, $3 < b$. We check for linear and colinear points.

*Variable* 1 has upper covers $\{a\}$. Candidate $1^\uparrow = a$: for every $x \geq 1$ in $A$, the only such $x$ is $a$, and $a \geq a$. So the condition $(\forall x \geq 1)\ x \geq a$ holds. Therefore **variable 1 is linear** with $1^\uparrow = a$.

*Variable* 3 has upper covers $\{b\}$. By the same argument, **variable 3 is linear** with $3^\uparrow = b$.

*Variable* 2 has upper covers $\{a, b\}$. For candidate $2^\uparrow = a$: we need $b \geq a$, but $b \not\geq a$. For candidate $2^\uparrow = b$: we need $a \geq b$, but $a \not\geq b$. So variable 2 is **not linear**.

No factor is colinear in the original poset (both $a$ and $b$ are binary factors, and neither scope is contained in the other).

We choose to retract variable 1.

**Step 1: Linear retraction of variable 1.**

**Factor-graph level.** Removing variable 1 from the poset gives $A' = \{2, 3, a, b\}$ with relations $2 < a$, $2 < b$, $3 < b$. Factor $a$ now has $\mathrm{ne}(a) = \{2\}$ (it has become unary). By the SP–B linear-retraction rule (Appendix D.1), the Hamiltonians on survivors are unchanged, so the surviving factor tables are:

$$\psi'_a(k_2) = \sum_{k_1 \in E_1} \psi_a(k_1, k_2) = \psi_a(1, k_2) + \psi_a(2, k_2), \qquad k_2 \in \{1, 2, 3\},$$
$$\psi'_b = \psi_b \quad \text{(unchanged)}.$$

The marginalization over $k_1$ arises because variable 1 had no other incident factors, so $P^{1\to a}$ was uniform at steady state (cf. equation (11)).

**CRN morphism $\mathcal{N}(r_1) = (\phi_0, \phi_1, \psi)$.**

$\phi_0$: **Delete** all bundles involving variable 1:
$$S^{a\to 1}, \quad P^{1\to a}.$$

This removes $3 + 3 = 6$ species: $S_0^{a\to 1}, S_1^{a\to 1}, S_2^{a\to 1}$ and $P_0^{1\to a}, P_1^{1\to a}, P_2^{1\to a}$.

$\phi_1$: **Delete** all reactions involving deleted species:

- Recycling of $S^{a\to 1}$: 2 reactions ($S_1^{a\to 1} \to S_0^{a\to 1}$, $S_2^{a\to 1} \to S_0^{a\to 1}$).
- Recycling of $P^{1\to a}$: 2 reactions ($P_1^{1\to a} \to P_0^{1\to a}$, $P_2^{1\to a} \to P_0^{1\to a}$).
- Sum-production of $S^{a\to 1}$: 6 reactions (the full $S^{a\to 1}$ table from Example E.1).
- Sum-production of $S^{a\to 2}$: all 6 reactions are deleted, because they use $P_{k_1}^{1\to a}$ as catalyst.
- Product-production of $P^{1\to a}$: 2 reactions ($P_0^{1\to a} \to P_1^{1\to a}$, $P_0^{1\to a} \to P_2^{1\to a}$).

Total deleted: $2 + 2 + 6 + 6 + 2 = \mathbf{18}$ reactions.

$\psi$: **Update** sum-production rates for $S^{a\to 2}$. After removing $P^{1\to a}$ from the catalytic set, the factor class $J_a$ now contains only the sum bundle $S^{a\to 2}$ (since $S^{a\to 1}$ was deleted), and $V_{J_a} = \{P^{2\to a}\}$. The sum bundle $S^{a\to 2}$ has $\mathrm{Cat}(S^{a\to 2}) = \varnothing$ (no remaining catalytic neighbors), and the factor table is now the unary table $\psi_a'(k_2) = \sum_{k_1} \psi_a(k_1, k_2)$. New sum-production reactions for $S^{a\to 2}$ (no catalysts, one reaction per state):

$$S_0^{a\to 2} \xrightarrow{\psi_a'(1)} S_1^{a\to 2},$$

$$S_0^{a\to 2} \xrightarrow{\psi_a'(2)} S_2^{a\to 2},$$

$$S_0^{a\to 2} \xrightarrow{\psi_a'(3)} S_3^{a\to 2},$$

where $\psi_a'(k_2) = \psi_a(1, k_2) + \psi_a(2, k_2)$. These replace the 6 deleted $S^{a\to 2}$ reactions with 3 new ones. All other surviving reactions (recycling of $S^{a\to 2}$, $S^{b\to 2}$, $S^{b\to 3}$, $P^{2\to a}$, $P^{2\to b}$, $P^{3\to b}$; sum-production of $S^{b\to 2}$, $S^{b\to 3}$; product-production of $P^{2\to a}$, $P^{2\to b}$, $P^{3\to b}$) retain their original rate constants.

**Intermediate CRN $\Gamma'$ after Step 1.**

| Bundle | Type | $|E_B|$ | Species |
|---|---|---|---|
| $S^{a\to 2}$ | sum | 3 | $S_0^{a\to 2}$, $S_1^{a\to 2}$, $S_2^{a\to 2}$, $S_3^{a\to 2}$ |
| $S^{b\to 2}$ | sum | 3 | $S_0^{b\to 2}$, $S_1^{b\to 2}$, $S_2^{b\to 2}$, $S_3^{b\to 2}$ |
| $S^{b\to 3}$ | sum | 2 | $S_0^{b\to 3}$, $S_1^{b\to 3}$, $S_2^{b\to 3}$ |
| $P^{2\to a}$ | product | 3 | $P_0^{2\to a}$, $P_1^{2\to a}$, $P_2^{2\to a}$, $P_3^{2\to a}$ |
| $P^{2\to b}$ | product | 3 | $P_0^{2\to b}$, $P_1^{2\to b}$, $P_2^{2\to b}$, $P_3^{2\to b}$ |
| $P^{3\to b}$ | product | 2 | $P_0^{3\to b}$, $P_1^{3\to b}$, $P_2^{3\to b}$ |

After Step 1: 22 species, $54 - 18 + 3 = 39$ reactions[1].

**Step 2: Colinear retraction of unary factor $a$.**

**Poset analysis of $A' = \{2, 3, a, b\}$.** After Step 1, factor $a$ is unary with $\mathrm{ne}(a) = \{2\}$. Its scope $\{2\}$ is contained in the scope $\{2, 3\}$ of factor $b$. Therefore **factor $a$ is colinear** with $a^{\downarrow} = 2$ (the unique lower cover of $a$).

We check the conditions of Lemma B.3. Let $B = A' \setminus \{a\} = \{2, 3, b\}$. In $B$, variable 2 has a single upper cover $b^{\uparrow} = b$. So variable 2 is linear in $B$, and Lemma B.3 applies.

**Factor-table update (Lemma B.3).** With $a^{\downarrow} = 2$ and $b^{\uparrow} = b$, the updated factor table is

$$\psi_b'(k_2, k_3) \;\propto\; \psi_b(k_2, k_3) \cdot \frac{\displaystyle\sum_{z \in E_a : z_{a\downarrow} = k_2} \psi_a(z)}{\psi_{a\downarrow}(k_2)}.$$

---

[1]We deleted 18 reactions and added 3 new sum-production reactions for the updated unary factor $\psi_a'$. Equivalently: $20 - 4 = 16$ recycling $+ 24 - 12 + 3 = 15$ sum-production $+ 10 - 2 = 8$ product-production $= 39$.

Since factor $a$ is already unary on variable 2 (i.e. $E_a = E_2$ and $z_{a\downarrow} = z = k_2$), the sum has a single term:

$$\sum_{z \in E_a : z_2 = k_2} \psi'_a(z) = \psi'_a(k_2).$$

Furthermore, the "variable region" $\psi_{a\downarrow}$ is absent here (variable 2 carries no separate unary factor in the current poset $A'$; equivalently, $\psi_2 \equiv 1$). So the update simplifies to

$$\psi'_b(k_2, k_3) = \psi_b(k_2, k_3) \cdot \psi'_a(k_2) = \psi_b(k_2, k_3) \cdot \big[\psi_a(1, k_2) + \psi_a(2, k_2)\big]. \tag{15}$$

This is still a $3 \times 2$ table (unchanged shape), but the entries have been modulated row-wise by the marginalized unary factor from variable 1.

**Reduced poset.** After removing factor $a$, the poset is $A'' = \{2, 3, b\}$ with relations $2 < b$, $3 < b$. This is the factor graph consisting of a single factor $b$ on variables $\{2, 3\}$ with the updated table $\psi'_b$.

**CRN morphism $\mathcal{N}(r_2) = (\phi_0, \phi_1, \psi)$.**

$\phi_0$: **Delete** all bundles involving factor $a$ (as sender or receiver):

$$S^{a \to 2}, \quad P^{2 \to a}.$$

This removes $4 + 4 = 8$ species.

$\phi_1$: **Delete** all reactions involving deleted species:

 – Recycling of $S^{a \to 2}$: 3 reactions.
 – Recycling of $P^{2 \to a}$: 3 reactions.
 – Sum-production of $S^{a \to 2}$: 3 reactions (the uncatalyzed reactions from Step 1).
 – Product-production of $P^{2 \to a}$: 3 reactions ($P_0^{2 \to a} + S_k^{b \to 2} \to \cdots$).
 – Product-production of $P^{2 \to b}$: These used $S_k^{a \to 2}$ as catalyst, so all 3 are deleted.

Total deleted: $3 + 3 + 3 + 3 + 3 = \mathbf{15}$ reactions.

$\psi$: **Update** rates.

*Sum-production of $S^{b \to 2}$:* Factor class $J_b$ now has $V_{J_b} = \{P^{2 \to b}, P^{3 \to b}\}$ (as before), but the table changes from $\psi_b$ to $\psi'_b$. The 6 reactions for $S^{b \to 2}$ retain their structure but their rates change from $\psi_b(k_2, k_3)$ to $\psi'_b(k_2, k_3)$:

| $(k_2, k_3)$ | Reaction | Updated rate |
|:---:|:---:|:---:|
| $(1, 1)$ | $S_0^{b \to 2} + P_1^{3 \to b} \to S_1^{b \to 2} + P_1^{3 \to b}$ | $\psi'_b(1, 1)$ |
| $(1, 2)$ | $S_0^{b \to 2} + P_2^{3 \to b} \to S_1^{b \to 2} + P_2^{3 \to b}$ | $\psi'_b(1, 2)$ |
| $(2, 1)$ | $S_0^{b \to 2} + P_1^{3 \to b} \to S_2^{b \to 2} + P_1^{3 \to b}$ | $\psi'_b(2, 1)$ |
| $(2, 2)$ | $S_0^{b \to 2} + P_2^{3 \to b} \to S_2^{b \to 2} + P_2^{3 \to b}$ | $\psi'_b(2, 2)$ |
| $(3, 1)$ | $S_0^{b \to 2} + P_1^{3 \to b} \to S_3^{b \to 2} + P_1^{3 \to b}$ | $\psi'_b(3, 1)$ |
| $(3, 2)$ | $S_0^{b \to 2} + P_2^{3 \to b} \to S_3^{b \to 2} + P_2^{3 \to b}$ | $\psi'_b(3, 2)$ |

*Sum-production of $S^{b \to 3}$:* Same structure, updated rates:

| $(k_2, k_3)$ | Reaction | Updated rate |
|---|---|---|
| $(1, 1)$ | $S_0^{b \to 3} + P_1^{2 \to b} \to S_1^{b \to 3} + P_1^{2 \to b}$ | $\psi_b'(1, 1)$ |
| $(2, 1)$ | $S_0^{b \to 3} + P_2^{2 \to b} \to S_1^{b \to 3} + P_2^{2 \to b}$ | $\psi_b'(2, 1)$ |
| $(3, 1)$ | $S_0^{b \to 3} + P_3^{2 \to b} \to S_1^{b \to 3} + P_3^{2 \to b}$ | $\psi_b'(3, 1)$ |
| $(1, 2)$ | $S_0^{b \to 3} + P_1^{2 \to b} \to S_2^{b \to 3} + P_1^{2 \to b}$ | $\psi_b'(1, 2)$ |
| $(2, 2)$ | $S_0^{b \to 3} + P_2^{2 \to b} \to S_2^{b \to 3} + P_2^{2 \to b}$ | $\psi_b'(2, 2)$ |
| $(3, 2)$ | $S_0^{b \to 3} + P_3^{2 \to b} \to S_2^{b \to 3} + P_3^{2 \to b}$ | $\psi_b'(3, 2)$ |

*Product-production of $P^{2 \to b}$*: After deletion of $S^{a \to 2}$, the catalytic set of $P^{2 \to b}$ becomes $\mathrm{Cat}(P^{2 \to b}) = \varnothing$ (variable 2 now touches only factor $b$). The product-production reactions become uncatalyzed:

$$P_0^{2 \to b} \xrightarrow{\kappa_{\mathrm{prod}}} P_1^{2 \to b}, \quad P_0^{2 \to b} \xrightarrow{\kappa_{\mathrm{prod}}} P_2^{2 \to b}, \quad P_0^{2 \to b} \xrightarrow{\kappa_{\mathrm{prod}}} P_3^{2 \to b}.$$

These replace the 3 deleted catalyzed reactions with 3 new uncatalyzed ones.

All other surviving reactions (recycling of $S^{b \to 2}$, $S^{b \to 3}$, $P^{2 \to b}$, $P^{3 \to b}$; product-production of $P^{3 \to b}$) are unchanged.

**Final reduced CRN $\Gamma'' = \mathcal{N}(A'', H'')$.** The reduced factor graph has variables $\{2, 3\}$ with state spaces $E_2 = \{1, 2, 3\}$, $E_3 = \{1, 2\}$, a single factor $b$ with $\mathrm{ne}(b) = \{2, 3\}$, and updated table $\psi_b'$ from equation (15). The compiled CRN has four bundles:

| Bundle | Type | $|E_B|$ | Species |
|---|---|---|---|
| $S^{b \to 2}$ | sum | 3 | $S_0^{b \to 2}$, $S_1^{b \to 2}$, $S_2^{b \to 2}$, $S_3^{b \to 2}$ |
| $S^{b \to 3}$ | sum | 2 | $S_0^{b \to 3}$, $S_1^{b \to 3}$, $S_2^{b \to 3}$ |
| $P^{2 \to b}$ | product | 3 | $P_0^{2 \to b}$, $P_1^{2 \to b}$, $P_2^{2 \to b}$, $P_3^{2 \to b}$ |
| $P^{3 \to b}$ | product | 2 | $P_0^{3 \to b}$, $P_1^{3 \to b}$, $P_2^{3 \to b}$ |

**Complete reaction list of $\Gamma''$.** *Recycling (8 reactions):*

$$S_1^{b \to 2} \xrightarrow{\kappa_r} S_0^{b \to 2}, \quad S_2^{b \to 2} \xrightarrow{\kappa_r} S_0^{b \to 2}, \quad S_3^{b \to 2} \xrightarrow{\kappa_r} S_0^{b \to 2},$$

$$S_1^{b \to 3} \xrightarrow{\kappa_r} S_0^{b \to 3}, \quad S_2^{b \to 3} \xrightarrow{\kappa_r} S_0^{b \to 3},$$

$$P_1^{2 \to b} \xrightarrow{\kappa_r} P_0^{2 \to b}, \quad P_2^{2 \to b} \xrightarrow{\kappa_r} P_0^{2 \to b}, \quad P_3^{2 \to b} \xrightarrow{\kappa_r} P_0^{2 \to b},$$

$$P_1^{3 \to b} \xrightarrow{\kappa_r} P_0^{3 \to b}, \quad P_2^{3 \to b} \xrightarrow{\kappa_r} P_0^{3 \to b}.$$

*Sum-production of $S^{b \to 2}$ (6 reactions, catalyst $P^{3 \to b}$):*

| $(k_2, k_3)$ | Reaction | Rate |
|---|---|---|
| $(1, 1)$ | $S_0^{b \to 2} + P_1^{3 \to b} \to S_1^{b \to 2} + P_1^{3 \to b}$ | $\psi_b'(1, 1)$ |
| $(1, 2)$ | $S_0^{b \to 2} + P_2^{3 \to b} \to S_1^{b \to 2} + P_2^{3 \to b}$ | $\psi_b'(1, 2)$ |
| $(2, 1)$ | $S_0^{b \to 2} + P_1^{3 \to b} \to S_2^{b \to 2} + P_1^{3 \to b}$ | $\psi_b'(2, 1)$ |
| $(2, 2)$ | $S_0^{b \to 2} + P_2^{3 \to b} \to S_2^{b \to 2} + P_2^{3 \to b}$ | $\psi_b'(2, 2)$ |
| $(3, 1)$ | $S_0^{b \to 2} + P_1^{3 \to b} \to S_3^{b \to 2} + P_1^{3 \to b}$ | $\psi_b'(3, 1)$ |
| $(3, 2)$ | $S_0^{b \to 2} + P_2^{3 \to b} \to S_3^{b \to 2} + P_2^{3 \to b}$ | $\psi_b'(3, 2)$ |

*Sum-production of $S^{b \to 3}$ (6 reactions, catalyst $P^{2 \to b}$):*

| $(k_2, k_3)$ | Reaction | Rate |
|---|---|---|
| $(1,1)$ | $S_0^{b\to3} + P_1^{2\to b} \to S_1^{b\to3} + P_1^{2\to b}$ | $\psi_b'(1,1)$ |
| $(2,1)$ | $S_0^{b\to3} + P_2^{2\to b} \to S_1^{b\to3} + P_2^{2\to b}$ | $\psi_b'(2,1)$ |
| $(3,1)$ | $S_0^{b\to3} + P_3^{2\to b} \to S_1^{b\to3} + P_3^{2\to b}$ | $\psi_b'(3,1)$ |
| $(1,2)$ | $S_0^{b\to3} + P_1^{2\to b} \to S_2^{b\to3} + P_1^{2\to b}$ | $\psi_b'(1,2)$ |
| $(2,2)$ | $S_0^{b\to3} + P_2^{2\to b} \to S_2^{b\to3} + P_2^{2\to b}$ | $\psi_b'(2,2)$ |
| $(3,2)$ | $S_0^{b\to3} + P_3^{2\to b} \to S_2^{b\to3} + P_3^{2\to b}$ | $\psi_b'(3,2)$ |

*Product-production of $P^{2\to b}$ (3 reactions, no catalysts):*

$$P_0^{2\to b} \xrightarrow{\kappa_{\text{prod}}} P_1^{2\to b}, \qquad P_0^{2\to b} \xrightarrow{\kappa_{\text{prod}}} P_2^{2\to b}, \qquad P_0^{2\to b} \xrightarrow{\kappa_{\text{prod}}} P_3^{2\to b}.$$

*Product-production of $P^{3\to b}$ (2 reactions, no catalysts):*

$$P_0^{3\to b} \xrightarrow{\kappa_{\text{prod}}} P_1^{3\to b}, \qquad P_0^{3\to b} \xrightarrow{\kappa_{\text{prod}}} P_2^{3\to b}.$$

**Reduced CRN summary.**

| | | |
|---|---|---|
| Species: | $4 + 3 + 4 + 3 = \mathbf{14}$ | (vs. 28 original, **50%** reduction) |
| Recycling reactions: | **10** | |
| Sum-bundle production: | $6 + 6 = \mathbf{12}$ | |
| Product-bundle production: | $3 + 2 = \mathbf{5}$ | |
| Total reactions: | **27** | (vs. 54 original, **50%** reduction) |

**Steady-state equations of the reduced CRN.** At a positive steady state of $\Gamma''$:

$$\frac{\kappa_r}{[S_0^{b\to2}]} [S_k^{b\to2}] = \sum_{k_3=1}^{2} \psi_b'(k, k_3) [P_{k_3}^{3\to b}], \qquad k \in \{1,2,3\}, \tag{16}$$

$$\frac{\kappa_r}{[S_0^{b\to3}]} [S_k^{b\to3}] = \sum_{k_2=1}^{3} \psi_b'(k_2, k) [P_{k_2}^{2\to b}], \qquad k \in \{1,2\}, \tag{17}$$

$$\frac{\kappa_r}{\kappa_{\text{prod}} [P_0^{2\to b}]} [P_k^{2\to b}] = 1, \qquad k \in \{1,2,3\}, \tag{18}$$

$$\frac{\kappa_r}{\kappa_{\text{prod}} [P_0^{3\to b}]} [P_k^{3\to b}] = 1, \qquad k \in \{1,2\}. \tag{19}$$

Equations (18) and (19) say that both product bundles are uniform at steady state (both variables now touch a single factor). The sum-bundle equations (16) and (17) implement BP on the single-factor graph $\{2,3,b\}$ with the updated table $\psi_b'$, which encodes the influence of the retracted variable 1 through the factor $\psi_a'(k_2) = \psi_a(1, k_2) + \psi_a(2, k_2)$ absorbed into $\psi_b'$.

**BP fixed-point preservation.** By Lemma 6.3, the marginals of variables 2 and 3 computed from the reduced CRN $\Gamma''$ coincide with those from the original CRN $\Gamma$. Concretely, at any positive steady state:

$$\left.\frac{[S_k^{b\to2}]}{\sum_{k'}[S_{k'}^{b\to2}]}\right|_{\Gamma''} = \left.\frac{[S_k^{b\to2}]}{\sum_{k'}[S_{k'}^{b\to2}]}\right|_{\Gamma}, \qquad k \in \{1,2,3\},$$

and similarly for the messages involving variable 3. The retracted variable 1 is no longer represented in the reduced CRN, but its effect on the surviving marginals is faithfully encoded in the updated rate constants $\psi_b'$.

**Remark (reduction on the chain graph is maximal).** The reduced poset $A'' = \{2, 3, b\}$ has no linear or colinear points: variable 2 has upper covers $\{b\}$ and variable 3 has upper covers $\{b\}$, but neither is linear because $b$ has two lower covers ($b^{\downarrow} = \{2, 3\}$, not a single element). Factor $b$ is binary, so it is not colinear. Therefore $A''$ is the *core* of the original poset $A$. By symmetry, retracting variable 3 first (instead of variable 1) would yield the same core $\{1, 2, a\}$ up to relabeling, with an analogous updated factor table $\psi'_a(k_1, k_2) = \psi_a(k_1, k_2) \cdot [\psi_b(k_2, 1) + \psi_b(k_2, 2)]$.

**Remark (Reconstructing beliefs on retracted variable nodes)** The retraction of a linear or colinear point induces a mapping that enables the reconstruction of beliefs for the retracted node from the node onto which it was retracted. By recursively applying this mapping, one can reconstruct the belief of any retracted node by following the path from a node in the reduced factor graph back to the node in question. This implies that from the steady-state concentrations of the reduced CRN, one can infer the concentrations of the original CRN. This reconstruction can be implemented using any CRN designed for filtering, such as Napp–Adams Belief Propagation (BP) CRNs (Napp & Adams, 2013), or specific CRNs proposed for the E-step of the Expectation-Maximization (EM) algorithm for learning Hidden Markov Models (HMMs) (Wiuf et al., 2023). Doing so is arguably less computationally expensive than executing full BP and it's associated CRN ODEs, particularly when only a few retracted belief nodes need to be reconstructed for downstream decision-making. A hypothetical example of this efficiency could arise when designing a synthetic chemotactic or phototactic cell. Such a cell might use a message-passing CRN to perform inference on sensory inputs, which then drive decision-making through competition between chemical concentrations, similar to the mechanisms described in (Tang et al., 2026). In this setting, retracted factor nodes that are not directly associated with observations possess factors that remain invariant to new sensory data. Consequently, the reconstruction mapping outlined above can be structurally enforced by design. These specific concentrations, which may play a critical role in the cell's decision-making, can thus be computed highly efficiently: first by performing the ODE associated to BP exclusively on the reduced CRN, and then by executing a targeted reconstruction only for the necessary nodes.

