# OpenReview forum: "Reduction of Probabilistic Chemical Reaction Networks"
_ICML.cc/2026/Conference — ICML 2026 regular_

### Official Review · Reviewer_Bp6A · 2026-03-06

**Soundness:** 3
**Presentation:** 2
**Significance:** 2
**Originality:** 3
**Overall Recommendation:** 4
**Confidence:** 1

**Summary:**

The manuscript studies how to reduce probabilistic chemical reaction networks (CRNs) that implement belief propagation (BP)-style inference without breaking the probabilistic semantics encoded by their steady states. The core contribution is a pipeline that compiles factor graphs into CRNs, reconstructs a factor graph from a CRN under explicit structural conditions (W1–W6), proves that positive steady states correspond to BP fixed points under an additional recycling condition (R1), and shows that factor-graph retractions can be transported to CRN reductions in a functorial way. Empirically, the paper reports substantial reductions in CRN size and simulation time on several synthetic graph families, while monitoring agreement of marginals on surviving variables.

**Compliance With Llm Reviewing Policy:**

Affirmed.

**Final Justification:**

I raised my rating from 3 to 4 because the author's rebuttal clarifies the scope and adds useful context to address my concerns. However, my evaluation is limited since the related topic is beyond my expertise.

**Key Questions For Authors:**

1. Could the authors provide a brief proposition, discussion, or table clarifying which classes of CRNs satisfy conditions W1–W6 and R1, which common constructions do not, and why?
2. Could the authors evaluate their method on non-binary variables and denser loopy structures, as well as compare it against classical CRN-reduction baselines? Demonstrating performance against these traditional approaches—even on a small subset of applicable tasks—would clearly show that this work solves a methodological gap rather than simply shifting the problem setting.

**Limitations:**

Yes

**Strengths And Weaknesses:**

Strengths:
- The paper is technically serious and its main claims are backed by a clear theorem chain: reconstruction of factor graphs from CRNs under (W1)–(W6), correspondence between positive steady states and BP fixed points under (R1), and transport of SP–B retractions through compilation.
- The empirical section also checks the right quantities, i.e., size reduction, simulation speed, and marginal agreement on surviving variables rather than only reporting runtime gains.
- The originality lies in the synthesis: the paper does not introduce a new CRN inference formalism from scratch, but it does give a novel bridge between Napp–Adams CRN compilation and SP–B factor-graph retractions, then shows that these reductions can be transported in a semantics-preserving way. That is a meaningful conceptual contribution.

Weaknesses:
- The theoretical results depend on fairly restrictive structural assumptions, especially (W1)–(W6) and (R1), so the guarantees apply to a specific compilable subclass rather than broadly to arbitrary biochemical CRNs. On the empirical side, all instances are synthetic and binary-valued, which limits how far one can generalize the practical claims.
- The formal sections are dense, with a large amount of notation introduced quickly. More importantly, the proofreading is not fully polished: the proof of Theorem 6.3 in p.20 appears duplicated, and there is a visible placeholder citation marker '(?)'.
- The potential impact seems specialized rather than broad at this stage. The paper makes a strong case for a niche but interesting setting, yet it does not show evidence on more realistic biochemical case studies, higher-cardinality variables, or direct comparisons with classical CRN reduction methods.
- While the combination is interesting, the work builds heavily on two existing ingredients: Napp–Adams compilation and SP–B retractions. The novelty is more in the formal integration and preservation results than in a fundamentally new inference or reduction paradigm. That is still valuable, but the paper should be careful not to imply a broader methodological novelty than it delivers.

---

> ### Author Rebuttal · Authors · 2026-03-30
>
> We thank the reviewer for the balanced and technically precise assessment.
>
> We agree with the reviewer's framing that the novelty lies in the formal integration and preservation results, and we will sharpen the paper's claim accordingly: to the best of our knowledge, this is the first semantics-preserving reduction method for BP-style compiled probabilistic CRNs.
>
> Q1 (Which CRNs satisfy W1--W6 and R1). We agree that this classification should be made more explicit. In the revision we will add a compact scope table to the appendix. Napp--Adams compiled CRNs satisfy W1--W6 and R1 by construction; CRNs obtained after our SP--B-induced reductions remain in this class by Proposition 6.2 / Lemma 6.3; arbitrary biochemical networks generally do not satisfy these conditions and are outside the present domain.
>
> Q2a (Non-binary variables and denser loopy structures). We ran $K \in \\{2, 3, 4, 5\\}$ sweeps across chain, tree, and loopy families. The reduction percentages are stable across all $K$, confirming that the reduction mechanism depends on graph topology rather than alphabet size. Chains: 10 to 1 variable, $94.0\\%$ species reduction for every $K$, with $(150, 200, 250, 300)$ species reducing to $(9, 12, 15, 18)$; BP discrepancy decreases from $7.93 \times 10^{-10}$ at $K=2$ to $5.55 \times 10^{-17}$ at $K=5$. Trees: 15 to 1 variable, $96.6\\%$ species reduction across all $K$. Loopy core+tendril: 16 to 4 variables, $68.2\\%$ species reduction across all $K$, with marginal error at machine precision for $K=4,5$. Mixed-cardinality chains (e.g. $[2,3,4,5,4,3,2]$) also reduce strongly: 7 to 1 variable, $94.0\\%$ species reduction, BP discrepancy $5.79 \times 10^{-13}$. We will include these results in the revision.
>
> On denser loopy structures specifically: a denser loopy graph corresponds to a larger irreducible core. The method removes attached tendrils and tree-like appendages, and then halts when no linear or colinear points remain. Accordingly, the reduction percentage decreases as the irreducible core occupies a larger fraction of the graph. The grid results already illustrate this limiting case: a grid has no reducible tendrils under the present moves and therefore shows $0%$ reduction. This is expected behavior rather than a failure of the method. The questions of denser loopy structure and larger cardinality are therefore separate, and our experiments address both.
>
> Q2b (Comparison to classical CRN-reduction baselines). We agree this is an important methodological question and we will make the answer more explicit in the revision. A direct baseline comparison is only informative when the baseline assumptions and preserved invariants align with the present objective, which is preservation of BP fixed points and inferred marginals on the surviving variables of a Napp--Adams compiled CRN.
>
> The central difficulty is that, in these compiled CRNs, message-passing semantics are carried by catalytic reactions. Catalyst species appear with zero net stoichiometry, contributing a zero column to the stoichiometric matrix $S$, so these dependencies are not visible to any method operating on $S$ or on graph-theoretic objects derived from it. This is the shared obstruction across all existing topological reduction methods, and it is structural rather than incidental.
>
> For the method of Hirono et al. [2], this means the candidate output-complete subnetworks satisfying $\lambda(\gamma) = 0$ collapse to trivial cases in this setting, so the method does not yield the nontrivial reductions considered here. For the Laplacian-based method of Gasparyan et al. [1], the same obstruction appears at the level of the complex graph: catalysts do not alter reaction complexes, so the inter-bundle dependencies carrying the BP semantics are absent from the Laplacian. These methods are therefore not natural baselines for the reduction problem studied in this paper. We will add this failure-mode argument explicitly in the revision, since it clarifies more directly than a high-level statement about invariants why an apples-to-apples numerical comparison is not available here.
>
> On the presentation issues: the duplicated proof of Theorem 6.3 is a proofreading error; the second version is the correct one and the first will be removed. The visible placeholder citation marker $(\text{?})$ will also be fixed. We will additionally add a pipeline figure and a 3-variable chain walkthrough to improve readability.

---

> > ### Author Rebuttal · Reviewer_Bp6A · 2026-04-02
> >
> > The rebuttal fully resolves my concerns and answers my questions. I have no further issue.

---

### Official Review · Reviewer_jH3L · 2026-03-09

**Soundness:** 3
**Presentation:** 2
**Significance:** 3
**Originality:** 3
**Overall Recommendation:** 4
**Confidence:** 2

**Summary:**

The paper studies chemical reaction network (CRN) design. CRNs have been shown to encode algorithms such as hidden Markov models, but often doing so results in very large networks that are too hard to implement physically. The paper proposes representing CRNs as factor graphs, and use that representation to reduce the size of the CRN. The paper identifies a subset of CRNs (specifically, those intended to implement Belief Propagation via the Napp-Adams construction), for which they can provide a mapping between the CRN representation and a factor graph. This mapping allows applying size reduction techniques to the CRN while preserving the steady-state marginal distribution of the selected variables. In the results, the authors show a reduction in the number of species of 97% in Chain and Tree factor graphs, followed by a 73% in Loopy graphs, 10% in Random graphs, and 0% in Grid graphs.

**Compliance With Llm Reviewing Policy:**

Affirmed.

**Final Justification:**

The authors clarified several points regarding the paper's contribution and scope in the rebuttal, which increases its significance and impact. My main concern about the presentation: the writing would benefit from extra clarity. However, in the rebuttal the authors acknowledged this and proposed improvements that I believe would improve readability.

**Key Questions For Authors:**

1. Could the authors provide more information about where is their approach situated in the state of the art? Is this paper the first that uses factor graphs to reduce the size of CRNs, or are there other similar techniques? That would help assess better the originality of the paper.

2. Does the approach provide a clear advantage against other CRN size reduction techniques?

3. Could the authors discuss whether the type of CRNs that satisfy the constraints of the method is broad? This is: whether this method could be applied in CRNs that are common for some applications. This would clarify the significance and areas of impact of the paper.

**Limitations:**

The paper is missing a Societal Impact section.

**Strengths And Weaknesses:**

Soundness: the submission seems technically sound. The paper exposes the relevant definitions, theorems, lemmas and equations to support the framework. The assumptions of their method are rather restrictive, but this is acknowledged in the discussed in the paper. The authors evaluate the framework across different graph topologies and discuss the observed results. However, the presentation makes it hard to follow the technical details and accurately verify the soundness (see below).

Presentation: the presentation is the main weakness of the paper. While the introduction is clear, the paper quickly becomes dense in the next section (2. Main Results), especially for readers who are not from the field. The paper would benefit from clarifying some of the background concepts (e.g. factor graphs, belief propagation, etc), and providing some examples to facilitate the reading. For instance, right after the introduction, in Sec. 2, the first paragraph explaining the claims of the paper is already very technical. Since a mathematical definition of these concepts is given later on (in the Background section) this section could benefit from using a simpler language, and an intuitive explanation and/or examples.

Overall, simplified explanations before the formal definitions, illustrative examples, and a clear position of the claims in the state of the art would greatly improve the paper readability.

Significance: the paper tackles an important problem. Implementing algorithms physically in CRNs is becoming increasingly possible, but the experimental size of the CRNs is still very constrained compared to the electronic counterparts. This work provides a way to reduce the size of CRNs, which would allow the implementation of complex algorithms with small networks that are experimentally feasible but conserve the main behavior of the algorithm.

Originality: The paper novelty lies in formalizing a pipeline that enables reducing the size of CRN by using the relationship between CRNs and formal graphs. While the idea that CRNs can implement probabilistic inference has been studied in previous work, the paper provides a deeper insight and pipeline for using this property to reduce CRN size. However, after my reading, I could not clearly situate this work in the state-of-the-art. It is not clear whether this is the first paper to use factor graphs to reduce the size of CRNs, or otherwise how it compares with similar techniques.

---

> ### Author Rebuttal · Authors · 2026-03-30
>
> We thank the reviewer for the careful technical reading and for clearly identifying presentation and positioning as the main issues.
>
> One correction first, which is important for interpreting the contribution. The paper does not "represent CRNs as factor graphs" in the forward direction. The primary direction is factor graph $\to$ CRN via Napp--Adams compilation. Theorem 4.7 is a recognition theorem for the inverse problem on a structured class: it identifies when a CRN already lies in the image of that compilation and reconstructs the factor graph encoded in its bundle and rate structure. We will make this more explicit earlier in the paper.
>
> Q1 (Position in the state of the art / originality). Prior work established that factor-graph message passing can be compiled into CRNs via the Napp--Adams construction. Our contribution adds three pieces that, to the best of our knowledge, were not previously available: (i) a recognition procedure recovering the source factor graph from a compiled CRN; (ii) a proof that positive steady states of such CRNs correspond to BP fixed points (Theorem 5.4); and (iii) a transport result showing that SP--B factor-graph reductions induce semantics-preserving CRN reductions through commutativity of reduction and compilation (Proposition 6.2 / Lemma 6.3). The novelty is therefore not that factor-graph inference can be implemented in CRNs, but that this compiled BP setting admits a reconstruction-and-reduction pipeline with preservation guarantees on the surviving variables.
>
> This has a concrete computational meaning in the loopy setting. When a desired marginal lies in a tendril removable by SP--B reduction, one may run BP on the irreducible core alone and recover the marginal by propagating outward using the probability kernels prescribed by the reduction, one kernel per edge. The biochemical system then need only encode inference on the core, with marginals along compressed branches recovered as needed. Such marginals, represented by species concentrations, could in principle be made available to downstream cellular control or decision-making mechanisms. We will illustrate this on examples in the revision.
>
> Q2 (Advantage relative to other CRN reduction techniques). The advantage is tied to the target invariant, and the reason existing methods cannot serve as baselines here is structural rather than incidental. In Napp--Adams compiled CRNs, the inferential semantics are carried by catalytic interactions. Catalyst species appear with zero net stoichiometry, contributing a zero column to the stoichiometric matrix $S$, so these dependencies are invisible to methods operating on $S$ or on graph-theoretic objects derived from it.
>
> For the structural reduction method of Hirono et al. [2], this means the candidate output-complete subnetworks satisfying $\lambda(\gamma) = 0$ collapse to trivial cases in this setting, so the method does not yield the nontrivial reductions studied here. For the Laplacian-based method of Gasparyan et al. [1], the same obstruction appears at the level of the complex graph: catalysts do not alter reaction complexes, so the inter-bundle dependencies carrying the BP semantics are absent from the Laplacian entirely. Neither class of method can detect the inferential structure it would need to preserve, which is precisely why a new reduction theory aligned with BP semantics is necessary. We will make this failure-mode argument explicit in the revision, since it more directly clarifies why the present reduction problem is not addressed by existing methods.
>
> Q3 (Breadth of the class satisfying the assumptions). The class is intentionally focused rather than broad: it consists of CRNs engineered to implement BP through Napp--Adams compilation. We do not claim that W1--W6 and R1 describe common biochemical CRNs in general. Rather, they identify the subclass for which graphical semantics can be reconstructed and then reduced in a semantics-preserving way. By Proposition 6.2, CRNs obtained after our reductions remain in this class, so the reduced systems are again valid compiled BP implementations rather than external approximations. We agree this should be stated more directly, and we will add both a scope paragraph and a compact summary table clarifying what lies inside and outside the present domain.
>
> On presentation, we fully agree with the reviewer's feedback. In the revision we will add a pipeline figure, a simple running example, a clearer positioning paragraph at the start of Section 2, and the missing Societal Impact section.
>
> [1]: Gasparyan, M., Bhalla, U. S., Radulescu, O., and Rao, S. Laplacian dynamics and kron reduction in species-reaction graphs of chemical reaction networks. bioRxiv, 2025. doi: 10.1101/2025.10.15.682662.
>
> [2]: Hirono, Y., Okada, T., Miyazaki, H., and Hidaka, Y. Structural reduction of chemical reaction networks based on topology. Nov 2021. PhysRevResearch.3.043123.

---

> > ### Author Rebuttal · Reviewer_jH3L · 2026-04-03
> >
> > Thank you for the thorough rebuttal. This clarifies the intended scope and novelty of the paper. I will raise my score to 4.

---

### Official Review · Reviewer_bZBU · 2026-03-12

**Soundness:** 3
**Presentation:** 2
**Significance:** 3
**Originality:** 4
**Overall Recommendation:** 4
**Confidence:** 1

**Summary:**

This paper uses graph factorization to reduce large chemical reaction network into smaller factorized networks while preserving the equilibrium point of the ODE systems. The paper uses belief propagation for probabilistic inference and shows how to recover the underlying factor graph structure from such CRNs using specific stoichiometric and rate conditions. They then apply factor-graph reductions (SP–B retractions) that remove variables or factors while updating neighboring factor tables so that belief propagation fixed points on the remaining variables are preserved.

**Compliance With Llm Reviewing Policy:**

Affirmed.

**Final Justification:**

The authors addressed most of the questions I have. I acknowledge that I have limited understanding of this paper, and the authors clarified many questions that I had. They explained how GRN could be reformulated into this framework, which is quite valuable. It would be great if the authors could include more benchmarking. However, it is difficult since not many methods have been developed for this task.

**Key Questions For Authors:**

1. Could reaction network be extended to concentration of mRNAs in order to study gene regulatory networks? If so, how would you imagine doing it in future work?
2. I am not that familiar with the field, but I wonder if the formulation of the ODEs in this paper can be generalized to enough chemical reaction systems out there in genomics or chemistry?
3. Another suggestion. Could you maybe compare to deep learning graph factorization method that don't intend to preserve fixed points of dynamical systems at all? Maybe the inductive bias from the dynamical system formulations helps you learn better pathways/gene modules/factors? It would be very interesting to see. A graph factorization method that doesn't perform well in genomics is DCDFG [1]. And one a well known paper for this task is in [2]. Could be a great improvement to your paper if you could show that you outperform them simply in gene module inference and evaluate in terms gene set enrichment analysis. Moreover, if you formulate the problem into biological pathway discovery, the paper would have a much larger audience and impact.
4. Would be nice if the authors could include experiments with real experiments rather than just simulated data.
5. Not enough discussion on how the method could be applied to real problems in chemistry and biology. Maybe add a paragraph in the discussion section.

I would be happy to increase score if questions are addressed.

[1] Lopez, R., Hütter, J.-C., Pritchard, J. K., & Regev, A. (2022). Large-Scale Differentiable Causal Discovery of Factor Graphs. Advances in Neural Information Processing Systems (NeurIPS).

[2] Segal, E., Shapira, M., Regev, A., Pe'er, D., Botstein, D., Koller, D., & Friedman, N. (2003). Module networks: identifying regulatory modules and their condition-specific regulators from gene expression data. Nature Genetics, 34(2), 166–176.

**Limitations:**

yes

**Strengths And Weaknesses:**

**Strength:**

Soundness:

This paper is technically sound.

Originality:

It introduces a brand new problem of graph factorization that preserves the fixed points of dynamical systems, which is something new to the field.

**Weakness:**

Presentation:
The presentation could be improved. For instance, there are too much math formulations in the main body. It would be great if a really simple to read workflow (figure) or description can be added to the main body.

Significance:
The paper has significance, but they could have given a more thorough description of a real application of their algorithm.

---

> ### Author Rebuttal · Authors · 2026-03-30
>
> We thank the reviewer for the encouraging assessment of the paper's originality and for the thoughtful suggestions about applications.
>
> We would first like to clarify one point in the summary. The paper is not a graph factorization or graph discovery method, and it does not infer graphical structure from data. The factor graph is reconstructed, not discovered, because the CRN already encodes it in its bundle and rate structure (Theorem 4.7). The contribution is a semantics-preserving reduction transport mechanism for BP-style compiled probabilistic CRNs.
>
> Q1--Q2 (Gene regulatory networks and generalizability). Our framework applies to any discrete factor graph compiled via Napp--Adams. Whether a given gene-regulatory ODE system can be re-expressed in this form is an open question that we cannot resolve here, and we do not want to overclaim direct applicability to existing genomics models. The more realistic near-term interpretation is synthetic or engineered molecular circuits deliberately designed to implement probabilistic computation in Napp--Adams form, to which our reduction would directly apply. We will add a discussion paragraph clarifying this intended domain and its current limitations.
>
> Q3 (Comparison to DCDFG / Module Networks). These methods solve a fundamentally different problem: they learn graphical structure from data, with no requirement to preserve dynamical fixed points of a compiled reaction system. Our method starts from a known compiled inference architecture and reduces it while preserving BP steady states exactly on the surviving variables. A direct numerical comparison would therefore conflate structure learning with semantics-preserving compiled-system reduction.
> That said, we agree there is an interesting future connection: one could use structure learning methods such as DCDFG or Module Networks upstream to obtain a candidate factor graph, then apply our reduction pipeline before biochemical implementation. In this sense our contribution is complementary as it provides the reduction step between a learned graphical model and a physically realizable CRN. We will present this explicitly as future work.
>
> Q4--Q5 (Real applications and discussion). The Napp--Adams construction is currently the only known method for implementing belief propagation inference in chemistry. It is already being applied to increasingly complex models: hidden Markov models, factor graph inference, active inference in synthetic cells. But it carries an unavoidable cost: the number of compiled chemical species grows rapidly with model complexity, quickly reaching sizes that are physically unrealizable. To the best of our knowledge, there was no principled way to address this bottleneck.
>
> Our paper provides the necessary tool to resolve this issue. The reduction pipeline collapses any Napp--Adams compiled CRN to its simplest possible correct implementation, its core, by removing all tree-like appendages that inflate network size without contributing new inferential content. This means every future system engineered via Napp--Adams for any application now has a principled path to physical realizability that did not exist before.
>
> To make the near term application concrete: a central goal in synthetic biology is engineering living cells that implement probabilistic computation at the molecular level, using CRNs as the biochemical substrate. Realizing such systems experimentally requires keeping the number of distinct molecular species within what can be feasibly engineered and maintained inside a cell, a constraint that compiled Napp--Adams networks routinely violate for even moderately complex inference tasks. Our reduction provides the missing step: before attempting physical realization, one reduces the compiled CRN to its minimal correct implementation, its core, discarding all chemical species that carry redundant inferential structure. The result is a network that implements the same BP inference with significantly fewer species, making experimental realization a realistic rather than a prohibitive goal. We will add a paragraph in the discussion section making this application pathway explicit. We will strengthen the discussion section to make this scope and motivation explicit.
>
> On the concern that the current experiments are binary-only, we now have additional sweeps over $K \in \\{ 2, 3, 4, 5 \\}$, including mixed-cardinality chains. Chain reduction remains $90.0\\%$ variable / $94.0\\%$ species for every $K$; tree reduction remains $93.3\\%$ / $96.6\\%$; loopy reduction remains $75.0\\%$ / $68.2\\%$, all with negligible marginal error. These results will be included in the revision

---

> > ### Author Rebuttal · Reviewer_bZBU · 2026-04-01
> >
> > The author fully resolved the questions I have. I am giving a score of 4.

---

### Official Review · Reviewer_JyTC · 2026-03-13

**Soundness:** 3
**Presentation:** 2
**Significance:** 3
**Originality:** 3
**Overall Recommendation:** 4
**Confidence:** 2

**Summary:**

In this paper, it is studied how to reduce probabilistic chemical reaction networks (CRNs) that implement probabilistic inference through belief propagation. The authors introduce a theoretical framework linking CRNs and factor graphs via a structure-preserving mapping. They first identify structural conditions under which a CRN can be interpreted as encoding a factor graph and provide a reconstruction procedure for recovering the graphical model. The authors show also that, under a specific condition, positive steady states of the CRN correspond to belief propagation fixed points on the reconstructed factor graph. The paper then demonstrates that a class of reductions for factor graphs, based on deformation retractions, can be transferred to the CRN setting. These reductions remove message bundles and adjust rate parameters while preserving the inference results associated with belief propagation, thanks to the ability of the compilation and reduction operators to commute with each other. The authors provide experiments showing that applying reductions before compiling the graphical model into a CRN can significantly reduce network size and accelerate ODE simulations.

**Compliance With Llm Reviewing Policy:**

Affirmed.

**Final Justification:**

Thanks for the rebuttal. Some aspects of the paper are now more clear. I do confirm my score.

**Key Questions For Authors:**

1.	Applicability of structural conditions.
How common are CRNs satisfying conditions (W1–W6) outside those generated through specific compilation procedures? Providing examples from existing CRN models would help clarify the scope of the method.
2.	Scaling with larger state spaces.
The experiments appear to use variables with two states. How does the effectiveness of the reduction and the size of the compiled CRN scale when variables have larger cardinalities?
3.	Characterizing reducible structures.
Can the authors provide a clearer characterization of the graph structures that benefit most from the reduction procedure?

**Limitations:**

yes

**Strengths And Weaknesses:**

Strengths
1. Strong theoretical structure
The paper presents a rigorous framework linking probabilistic graphical models and chemical reaction networks. The reconstruction theorem identifying when a CRN encodes a factor graph and the proof relating steady states of the CRN to belief propagation fixed points provide a clear and well-motivated theoretical foundation. The framework is further strengthened by additional structural results, such as the formalization of the compilation process and the observation that reduction and compilation commute, which clarifies how reductions in the graphical model translate to reductions in the corresponding CRN.
2. Correctness guarantees
A key strength of the approach is that the reductions preserve belief propagation fixed points. This ensures that the reduced CRN maintains the same inference results on the remaining variables, providing a strong theoretical guarantee.
3. Clear motivation
The paper addresses the challenge that CRN implementations of probabilistic inference can grow very large and become difficult to simulate. Providing a principled way to simplify such systems while maintaining their inference behavior is an important goal.


Weaknesses
1. Restrictive structural assumptions
The theoretical framework relies on a set of structural conditions (W1–W6) that define when a CRN can be interpreted as representing a factor graph. While these conditions are clearly stated, it is not entirely clear how frequently they occur outside networks produced by specific compilation procedures. A clearer discussion of the practical scope of these assumptions would strengthen the work.
2. Limited experimental evaluation
The empirical study mainly considers synthetic factor graphs with binary variables. Although the results demonstrate substantial reductions in CRN size and simulation time in certain cases, the experimental setting is somewhat narrow. For example:
•	All variables have only two states.
•	The graphs are synthetically generated.
This makes it harder to evaluate how well the approach performs on more realistic systems.
3. Density of technical sections
Although the paper is generally well organized, the formal sections introduce many definitions and conditions in a relatively short space, making the paper hard to read. Additional examples or intuitive explanations of the structural conditions and reduction mechanism could improve clarity and accessibility.

---

> ### Author Rebuttal · Authors · 2026-03-30
>
> We thank the reviewer for the thoughtful reading and for the precise questions on scope, cardinality, and reducible structure.
> One framing point first. The paper does not represent arbitrary CRNs as factor graphs. It studies CRNs arising from Napp--Adams-style BP compilation, recovers the graphical semantics already encoded in their bundle structure, and uses commutativity of reduction and compilation to transport SP--B factor-graph reductions into semantics-preserving CRN reductions. To the best of our knowledge, this is the first semantics-preserving reduction method for BP-style compiled probabilistic CRNs.
>
> Q1 (Applicability of W1--W6). Conditions W1--W6 characterize the Napp--Adams-compiled class studied in the paper. Any CRN engineered via this construction satisfies them by design, for any factor graph over any finite alphabet. So this is a recognition-and-reduction theorem for a structured subclass, not a universality claim about arbitrary biochemical networks. We agree that the scope should be clarified more explicitly. In the revision, we will add a short scope paragraph and a compact appendix table distinguishing:
>
>  (i) Napp--Adams compiled CRNs, which satisfy W1--W6 and R1 by construction.
>
> (ii) CRNs obtained after our reductions, which remain in the same class by Proposition 6.2
>
> (iii) General biochemical CRNs, which need not satisfy these conditions.
>
> Q2 (Scaling with larger state spaces). We ran cardinality sweeps for $K \in {2, 3, 4, 5}$ across chain, tree, and loopy families. The reduction percentages are stable across all $K$, consistent with the retraction conditions depending on graph topology rather than alphabet size. Concretely: 10-node chains reduce from $(150, 200, 250, 300)$ species to $(9, 12, 15, 18)$, giving $90.0\\%$ variable and $94.0\\%$ species reduction in every case, with BP discrepancy decreasing from $7.93 \times 10^{-10}$ at $K=2$ to $5.55 \times 10^{-17}$ at $K=5$. Depth-4 trees reduce 15 to 1 variable with $96.6\\%$ species reduction across all $K$. Loopy core+tendril graphs reduce 16 to 4 variables with $68.2\\%$ species reduction across all $K$, with marginal error at machine precision for $K=4,5$. We also ran mixed-cardinality chains (e.g. $[2,3,4,5,4,3,2]$): 7 to 1 variable, $94.0\\%$ species reduction, BP discrepancy $5.79 \times 10^{-13}$. We will include these results in the revision.
>
> Q3 (Which structures are reducible). The structures that benefit most are those with tree-like appendages -- tendrils -- attached to a smaller irreducible core. SP--B reductions systematically remove all such tendrils, collapsing the factor graph to its $\textit{core}$: the minimal subgraph from which no further reduction is possible without altering BP fixed points. More precisely, SP--B reductions eliminate variables that are linear points in the factor-graph poset, i.e. variables whose scope is contained in the scope of a neighboring factor. This covers all non-endpoint variables in chains, internal variables in trees, and tendril variables attached to loopy cores.
>
> The core is the irreducible inferential content of the model. Everything outside it is redundant structure that inflates the compiled CRN without contributing new information. The large reductions on chains and trees reflect that these graphs are almost entirely tendril; their cores are trivial. The partial reductions on loopy-core graphs reflect that the loopy core must be preserved. Grid graphs have no tendrils at all and reduce to $0\\%$, which is the method correctly identifying that the entire graph is its own core. This last case serves as a control: a method that claimed to reduce grids would be destroying inferential content, not simplifying it.
> Importantly, the questions of core density and state cardinality are independent. The cardinality sweeps show that reducibility depends on topology, not alphabet size. The grid and dense loopy results show that the method correctly halts when the core is reached, regardless of how dense that core is.
>
> On presentation, we agree with the reviewer's comment. We will add a pipeline figure and a 3-variable chain walkthrough so that the recognition, BP correspondence, and reduction transport can be read once concretely before the full formalism.

---

> > ### Author Rebuttal · Reviewer_JyTC · 2026-04-02
> >
> > The rebuttal is satisfactory.

---

### Decision · Program_Chairs · 2026-04-30

**Decision:**

Accept (regular)

**Comment:**

The paper studies how to reduce probabilistic chemical reaction networks (CRNs) that implement probabilistic inference through belief propagation. It links the CRNs to factor graphs, proves correspondence between positive steady states and belief-propagation fixed points, and maps factor-graph reductions back to the CRN to preserve inference while shrinking the network.

The reviews were mildly positive but they all consider the contribution borderline. On the positive side, reviewers appreciated the theoretical rigor in proving the connection between CRNs and factor graphs. Some concerns have been expressed to better clarify the paper’s precise contribution, which seem to have been solved during the rebuttal. However, concerns about restrictive structural assumptions and narrow experiments on synthetic binary graphs remain.

Overall, I agree with the reviewers that this paper is borderline: it is very strong on a theoretical standpoint, but lacks concrete empirical validation beyond synthetic datasets to make it practical. Moreover, the results are only applicable to the subset of Napp-Adams compiled CRNs, which may restrict its applicability.